# IRIS: LLM-ASSISTED STATIC ANALYSIS FOR DETECTING SECURITY VULNERABILITIES

**Ziyang Li**
University of Pennsylvania
liby99@cis.upenn.edu

**Saikat Dutta**
Cornell University
saikatd@cornell.edu

**Mayur Naik**
University of Pennsylvania
mhnaik@cis.upenn.edu

## ABSTRACT

Software is prone to security vulnerabilities. Program analysis tools to detect them have limited effectiveness in practice due to their reliance on human labeled specifications. Large language models (or LLMs) have shown impressive code generation capabilities but they cannot do complex reasoning over code to detect such vulnerabilities especially since this task requires whole-repository analysis. We propose IRIS, a neuro-symbolic approach that systematically combines LLMs with static analysis to perform whole-repository reasoning for security vulnerability detection. Specifically, IRIS leverages LLMs to infer taint specifications and perform contextual analysis, alleviating needs for human specifications and inspection. For evaluation, we curate a new dataset, CWE-Bench-Java, comprising 120 manually validated security vulnerabilities in real-world Java projects. A state-of-the-art static analysis tool CodeQL detects only 27 of these vulnerabilities whereas IRIS with GPT-4 detects 55 (+28) and improves upon CodeQL's average false discovery rate by 5% points. Furthermore, IRIS identifies 4 previously unknown vulnerabilities which cannot be found by existing tools. IRIS is available publicly at https://github.com/iris-sast/iris.

## 1 INTRODUCTION

Security vulnerabilities pose a major threat to the safety of software applications and its users. In 2023 alone, more than 29,000 CVEs were reported—almost 4000 higher than in 2022 (CVE Trends). Detecting vulnerabilities is extremely challenging despite advances in techniques to uncover them. A promising such technique called static taint analysis is widely used in popular tools such as GitHub CodeQL (Avgustinov et al., 2016), Facebook Infer (FB Infer), Checker Framework (Checker Framework), and Snyk Code (Snyk.io). These tools, however, face several challenges that greatly limit their effectiveness and accessibility in practice.

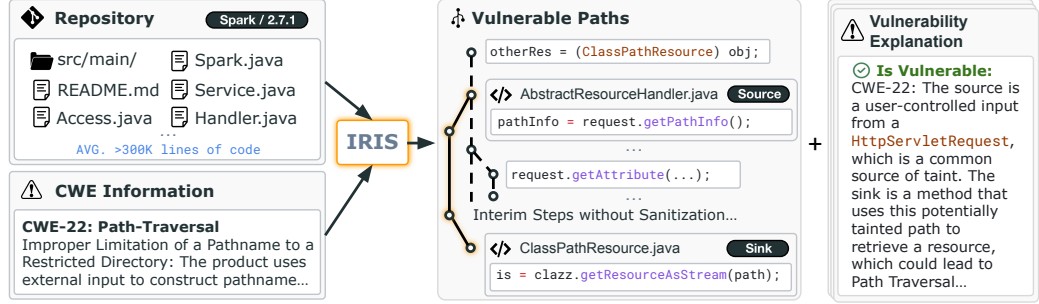

Figure 1: Overview of the IRIS neuro-symbolic system. It checks a given whole repository for a given type of vulnerability (CWE) and outputs a set of potential vulnerable paths with explanations.

**False negatives due to missing taint specifications of third-party library APIs.** First, static taint analysis predominantly relies on *specifications* of third-party library APIs as sources, sinks, or sanitizers. In practice, developers and analysis engineers have to manually craft such specifications based on their domain knowledge and API documentation. This is a laborious and error-prone process that often leads to missing specifications and incomplete analysis of vulnerabilities. Further,

even if such specifications may exist for many libraries, they need to be periodically updated to capture changes in newer versions of such libraries and also cover new libraries that are developed.

**False positives due to lack of precise context-sensitive and intuitive reasoning.** Second, it is well-known that static analysis often suffers from low precision, i.e., it may generate many false alarms (Kang et al., 2022; Johnson et al., 2013). Such imprecision stems from multiple sources. For instance, the source or sink specifications may be spurious, or the analysis may over-approximate over branches in code or possible inputs. Further, even if the specifications are correct, the context in which the detected source or sink is used may not be exploitable. Hence, a developer may need to triage through several potentially false security alerts, wasting significant time and effort.

**Limitations of prior data-driven approaches to improve static taint analysis.** Many techniques have been proposed to address the challenges of static taint analysis. For instance, Livshits et al. (2009) proposed a probabilistic approach, MERLIN, to automatically mine taint specifications. A more recent work, Seldon (Chibotaru et al., 2019), improves the scalability of this approach by formulating the taint specification inference problem as a linear optimization task. However, such approaches rely on analyzing the code of third-party libraries to extract specifications, which is expensive and hard to scale. Researchers have also developed statistical and learning-based techniques to mitigate false positive alerts (Jung et al., 2005; Heckman & Williams, 2009; Hanam et al., 2014). However, such approaches still have limited effectiveness in practice (Kang et al., 2022).

Large Language Models (or LLMs) have made impressive strides in code generation and summarization. LLMs have also been applied to code related tasks such as program repair (Xia et al., 2023), code translation (Pan et al., 2024), test generation (Lemieux et al., 2023), and static analysis (Li et al., 2024). Recent studies (Steenhoek et al., 2024; Khare et al., 2023) evaluated LLMs' effectiveness at detecting vulnerabilities at the method level and showed that LLMs fail to do complex reasoning with code, especially because it depends on the *context* in which the method is used in the project. On the other hand, recent benchmarks like SWE-Bench (Jimenez et al., 2023) show that LLMs are also poor at doing project-level reasoning. Hence, an intriguing question is whether LLMs can be combined with static analysis to improve their reasoning capabilities. In this work, we answer this question in the context of vulnerability detection and make the following contributions:

**Approach.** We propose **IRIS**, a neuro-symbolic approach for vulnerability detection that combines the strengths of static analysis and LLMs. Fig. 1 presents an overview of IRIS. Given a project to analyze for a given vulnerability class (or CWE), IRIS applies LLMs for mining CWE-specific taint specifications. IRIS augments such specifications with CodeQL, a tool for static taint analysis. Our intuition here is because LLMs have seen numerous usages of such library APIs, they have an understanding of the relevant APIs for different CWEs. Further, to address the imprecision problem of static analysis, we propose a contextual analysis technique with LLMs that reduces the false positive alarms and minimizes the triaging effort for developers. Our key insight is that encoding the code-context and path-sensitive information in the prompt elicits more reliable reasoning from LLMs. Finally, our neuro-symbolic approach allows LLMs to do more precise whole-repository reasoning and minimizes the human effort involved in using static analysis tools.

**Dataset.** We curate a dataset of manually vetted and compilable Java projects, **CWE-Bench-Java**, containing 120 vulnerabilities (one per project) across four common vulnerability classes. The projects in the dataset are complex, containing 300K lines of code on average, and 10 projects with more than a million lines of code each, making it a challenging benchmark for vulnerability detection. The dataset and the corresponding scripts to fetch, build, and analyze the Java projects are available publicly at `https://github.com/iris-sast/cwe-bench-java`.

**Results.** We evaluate IRIS on CWE-Bench-Java using 7 diverse open- and closed-source LLMs. Overall, IRIS obtains the best results with GPT-4, detecting 55 vulnerabilities, which is 28 (103.7%) more than CodeQL, the existing best-performing static analyzer. We show that the increase is not at the expense of false positives, as IRIS with GPT-4 achieves an average false discovery rate of 84.82%, which is 5.21% lower than that of CodeQL. Further, when applied to the latest versions of 30 Java projects, IRIS with GPT-4 discovered 4 previously unknown vulnerabilities.

Figure 2: An example of Code Injection (CWE-94) vulnerability found in cron-utils (CVE-2021-41269) that CodeQL fails to detect. We number the program points of the vulnerable path.

## 2  MOTIVATING EXAMPLE

We illustrate the effectiveness of IRIS in detecting a previously known code-injection (CWE-094) vulnerability in cron-utils (ver. 9.1.5), a Java library for Cron data manipulation. Fig. 2 shows the relevant code snippets. A user-controlled string `value` passed into `isValid` function is transferred without sanitization to the `parse` function. If an exception is thrown, the function constructs an error message with the input. However, the error message is used to invoke method `buildConstraintViolationWithTemplate` of class `ConstraintValidatorContext` in `javax.validator`, which interprets the message string as a Java Expression Language (Java EL) expression. A malicious user may exploit this vulnerability by crafting a string containing a shell command such as `Runtime.exec('rm -rf /')` to delete critical files on the server.

Detecting this vulnerability poses several challenges. First, the cron-utils library consists of 13K SLOC (lines of code excluding blanks and comments), which needs to be analyzed to find this vulnerability. This process requires analyzing data and control flow across several internal methods and third-party APIs. Second, the analysis needs to identify relevant *sources* and *sinks*. In this case, the `value` parameter of the public `isValid` method may contain arbitrary strings when invoked, and hence may be a source of malicious data. Additionally, external APIs like `buildConstraintViolationWithTemplate` can execute arbitrary Java EL expressions, hence they should be treated as sinks that are vulnerable to Code Injection attacks. Finally, the analysis also requires identifying any sanitizers that block the flow of untrusted data.

Modern static analysis tools, like CodeQL, are effective at tracing taint data flows across complex codebases. However, CodeQL fails to detect this vulnerability due to missing specifications. CodeQL includes many manually curated specifications for sources and sinks across more than 360 popular Java library modules. However, manually obtaining such specifications requires significant human effort to analyze, specify, and validate. Further, even with perfect specifications, CodeQL may often generate numerous false positives due to a lack of contextual reasoning, increasing the developer's burden of triaging the results.

In contrast, IRIS takes a different approach by inferring project- and vulnerability-specific specifications *on-the-fly* by using LLMs. The LLM-based components in IRIS correctly identify the untrusted source and the vulnerable sink. IRIS augments CodeQL with these specifications and successfully detects the unsanitized dataflow path between the detected source and sink in the repository. However, augmented CodeQL produces many false positives, which are hard to eliminate using logical rules. To solve this challenge, IRIS encodes the detected code paths and the surrounding context into a simple prompt and uses an LLM to classify it as true or false positive. Specifically, out of 8 paths reported by static analysis, 5 false positives are filtered out, leaving the path in Fig. 2 as one of the final alarms. Overall, we observe that IRIS can detect many such vulnerabilities that are beyond the reach of CodeQL-like static analysis tools, while keeping false alarms to a minimum.

## 3  IRIS FRAMEWORK

At a high level, IRIS takes a Java project $P$, the vulnerability class $C$ to detect, and a large language model `LLM`, as inputs. IRIS statically analyzes the project $P$, checks for vulnerabilities specific to $C$, and returns a set of potential security alerts $A$. Each alert is accompanied by a unique code path from a taint source to a taint sink that is vulnerable to $C$ (i.e., the path is unsanitized).

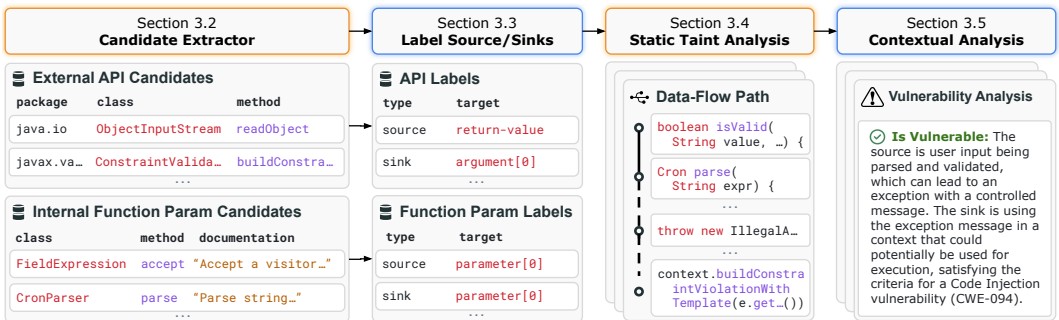

Figure 3: An illustration of the IRIS pipeline.

As illustrated in Fig. 3, IRIS has four main stages: First, IRIS builds the given Java project and uses static analysis to extract all candidate APIs, including invoked external APIs and internal function parameters. Second, IRIS queries an LLM to label these APIs as sources or sinks that are specific to the given vulnerability class $C$. Third, IRIS transforms the labeled sources and sinks into specifications that can be fed into a static analysis engine, such as CodeQL, and runs a vulnerability class-specific taint analysis query to detect vulnerabilities of that class in the project. This step generates a set of vulnerable code paths (or alerts) in the project. Finally, IRIS triages the generated alerts by automatically filtering false positives, and presents them to the developer.

## 3.1 PROBLEM STATEMENT

We formally define the static taint analysis problem for vulnerability detection. Given a project $P$, taint analysis extracts an inter-procedural data flow graph $\mathbb{G} = (\mathbb{V}, \mathbb{E})$, where $\mathbb{V}$ is the set of nodes representing program expressions and statements, and $\mathbb{E} \subseteq \mathbb{V} \times \mathbb{V}$ is the set of edges representing data or control flow edges between the nodes. A vulnerability detection task comes with two sets $\boldsymbol{V}_{source}^{C} \subseteq \mathbb{V}$, $\boldsymbol{V}_{sink}^{C} \subseteq \mathbb{V}$ that denote source nodes where tainted data may originate and sink nodes where a security vulnerability can occur if tainted data reaches it, respectively. Naturally, different classes $C$ of vulnerabilities (or CWEs) have different source and sink specifications. Additionally, there can be sanitizer specifications, $\boldsymbol{V}_{sanitizer}^{C} \in \mathbb{V}$, that block the flow of tainted data (such as escaping special characters in strings).

The goal of taint analysis is to find pairs of sources and sinks, $(V_s \in \boldsymbol{V}_{source}^{C}, V_t \in \boldsymbol{V}_{sink}^{C})$, such that there is an *unsanitized* path from the source to the sink. More formally, $Unsanitized\_Paths(V_s, V_t) = \exists\, Path(V_s, V_t)$ s.t. $\forall V_n \in Path(V_s, V_t), V_n \notin \boldsymbol{V}_{sanitizer}^{C}$. Here, $Path(V_1, V_k)$ denotes a sequence of nodes $(V_1, V_2, \dots, V_k)$, such that $V_i \in \mathbb{V}$ and $\forall i \in 1\ to\ k-1 : (v_i, v_{i+1}) \in \mathbb{E}$.

Two key challenges in taint analysis include: 1) identifying relevant taint specifications for each class C that can be mapped to $\boldsymbol{V}_{source}^{C}$, $\boldsymbol{V}_{sink}^{C}$ for any project $P$, and 2) effectively eliminating false positive paths in $Unsanitized\_Paths(V_s, V_t)$ identified by taint analysis. In the following sections, we discuss how we address each challenge by leveraging LLMs.

## 3.2 CANDIDATE SOURCE/SINK EXTRACTION

A project may use various third-party APIs whose specifications may be unknown—reducing the effectiveness of taint analysis. In addition, internal APIs might accept untrusted input from downstream libraries. Hence, our goal is to automatically infer specifications for such APIs. We define a specification $S^C$ as a 3-tuple $\langle T, F, R \rangle$, where $T \in \{ReturnValue, Argument, Parameter, \dots\}$ is the type of node to match in $\mathbb{G}$, $F$ is an N-tuple of strings describing the package, class, method name, signature, and argument/parameter position (if applicable) of an API, and $R \in \{Source, Sink, Taint\text{-}Propagator, Sanitizer\}$ is the role of the API. For example, the specification $\langle Argument, (\texttt{java.lang}, \texttt{Runtime}, \texttt{exec}, (\texttt{String[]}), 0), Sink \rangle$ denotes that the first argument of $\texttt{exec}$ method of $\texttt{Runtime}$ class is a sink for a vulnerability class (OS command injection). A static analysis tool maps these specifications to sets of nodes $\boldsymbol{V}_{source}^{C}$ or $\boldsymbol{V}_{sink}^{C}$ in $\mathbb{G}$.

To identify taint specifications $\boldsymbol{S}_{source}^{C}$ and $\boldsymbol{S}_{sink}^{C}$, we first extract $\boldsymbol{S}^{\text{ext}}$: external library APIs that are invoked in the given Java project and are potential candidates to be taint sources or sinks. We also extract $\boldsymbol{S}^{\text{int}}$, internal library APIs that are public and may be invoked by a downstream library. We

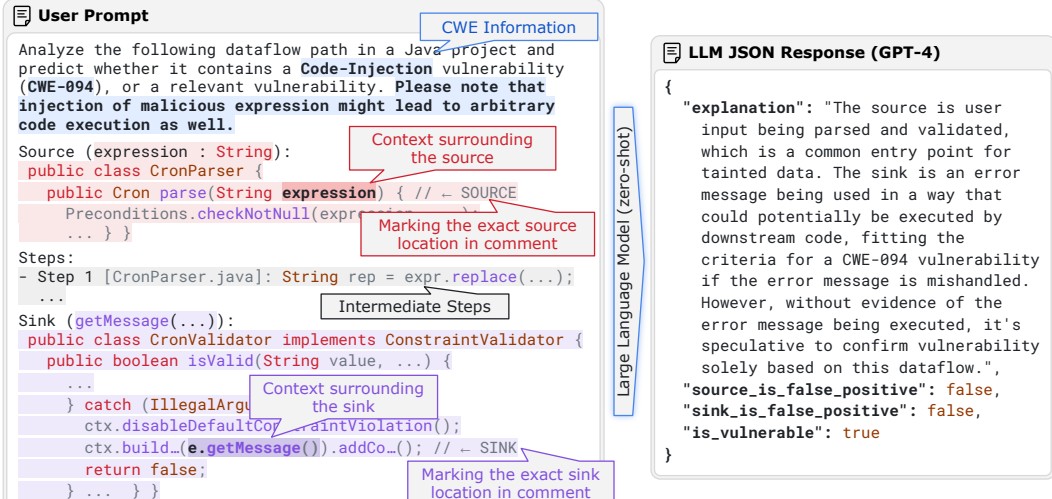

Figure 4: LLM user prompt and response for contextual analysis of dataflow paths. In the user prompt, we mark with color the CWE and path information that is filling the prompt template. For cleaner presentation, we modify the snippets and left out the system prompt.

use CodeQL to extract such candidates and their corresponding metadata such as method name, type signature, enclosing packages and classes, and even JavaDoc documentations, if applicable.

### 3.3 INFERRING TAINT SPECIFICATIONS USING LLMS

We develop an automated specification inference technique: $LabelSpecs(\boldsymbol{S}^{\#}, \text{LLM}, C, R) = \boldsymbol{S}_R^C$, where $\boldsymbol{S}^{\#} = \boldsymbol{S}^{\text{ext}} \cup \boldsymbol{S}^{\text{int}}$ are candidate specifications for sources and sinks. In this work, we do not consider sanitizer specifications, because they typically do not vary for the vulnerability classes that we consider. We use LLMs to infer taint specifications. Specifically, external APIs in $\boldsymbol{S}^{\text{ext}}$ can be classified as either source or sink, while internal APIs in $\boldsymbol{S}^{\text{int}}$ can have their formal parameters identified as sources. In the Appendix, we show the user prompts for inferring source and sink specifications from external APIs and internal function formal parameters.

Due to the sheer number of APIs to be labeled, we insert a batch of APIs in a single prompt and ask the LLM to respond with JSON formatted strings. The batch size is a tunable hyper-parameter. We adopt few-shot (usually 3-shot) prompting strategy for labeling external APIs, while zero-shot is used for labeling internal APIs. Notably for internal APIs, we also include information from repository readme and JavaDoc documentations, if applicable. In practice, we find that this extra information helps LLM understand the high-level purpose and usage of the codebase, resulting in better labeling accuracy. At the end of this stage, we have successfully obtained $\boldsymbol{S}_{source}^C$ and $\boldsymbol{S}_{sink}^C$ which are going to be used by the static analysis engine in the next stage.

### 3.4 VULNERABILITY DETECTION

Once we obtain all the source and sink specifications from the LLM, the next step is to combine it with a static analysis engine to detect vulnerable paths, i.e., $Unsanitized\_Paths(V_s, V_t)$, in a given project. In this work, we use CodeQL (GitHub, 2024a) for this step. CodeQL represents programs as data flow graphs and provides a query language, akin to Datalog (Smaragdakis & Bravenboer, 2010), to analyze such graphs. Many security vulnerabilities can be modeled using *queries* written in CodeQL and can be executed against data flow graphs extracted from such programs. Given a data flow graph $\mathbb{G}^P$ of a project $P$, CWE-specific source and sink specifications, and a query for a given vulnerability class $C$, CodeQL returns a set of unsanitized paths in the program. Formally,

$$CodeQL(\mathbb{G}^P, \boldsymbol{S}_{source}^C, \boldsymbol{S}_{sink}^C, Query^C) = \{Path_1, \ldots, Path_k\}.$$

CodeQL itself contains numerous specifications of third-party APIs for each vulnerability class. However, as we show later in our evaluation, despite having such specialized queries and extensive specifications, CodeQL fails to detect a majority of vulnerabilities in real-world projects. For our

analysis, we write a specialized CodeQL query for each vulnerability that uses our mined specifications instead of those provided by CodeQL. Details of our queries are described in the Appendix.

## 3.5 Triaging of Alerts via Contextual Analysis

Inferring taint specifications only solves part of the challenge. We observe that while LLMs help uncover many new API specifications, sometimes they detect specifications that are not relevant to the vulnerability class being considered, resulting in too many predicted sources or sinks and consequently many spurious alerts as a result. For context, even a few hundred taint specifications may sometimes produce thousands of $Unsanitized\_Paths(V_s, V_t)$ that a developer needs to triage. To reduce the developer burden, we also develop an LLM-based filtering method, $FilterPath(Path, \mathbb{G}, \text{LLM}, C) = \text{True}|\text{False}$ that classifies a detected vulnerable path ($Path$) in $\mathbb{G}$ as a true or false positive by leveraging context-based and natural language information.

Fig. 4 presents an example prompt for contextual analysis. The prompt includes CWE information and code snippets for nodes along the path, with an emphasis on the source and sink. For the intermediate steps, we include the file names and the line of code. When the path is too long, we keep only a subset of nodes to limit the size of the prompt. More details and design decisions of this prompt are described in the Appendix. We expect the LLM to respond in JSON format with the final verdict as well as an explanation to the verdict. The JSON format prompts the LLM to generate the explanation before delivering the final verdict, as presenting the judgment after the reasoning process is known to yield better results. In addition, if the verdict is false, we ask the LLM to indicate whether the source or sink is a false positive, which helps to prune other paths and thereby save on the number of calls to the LLM.

## 3.6 Evaluation Metrics

We evaluate the performance of IRIS and its baselines using three key metrics: number of vulnerability detected (*#Detected*), average false discovery rate (*AvgFDR*), and average F1 (*AvgF1*). For evaluation, we assume that we have a dataset $\mathcal{D} = \{P_1, \ldots, P_n\}$ where each $P_i$ is a Java project, and known to contain at least one vulnerability. The label for a project $P$ is provided as a set of crucial program points $\mathbf{V}_{\text{vul}}^P = \{V_1, \ldots, V_n\}$ where the vulnerable paths should pass through, indicated by $Path \cap \mathbf{V}_{\text{vul}}^P \neq \emptyset$. In practice, these are typically the patched methods that can be collected from each vulnerability report. If at least one detected vulnerable path passes through a fixed location for the given vulnerability, then we consider the vulnerability detected. Let $Paths^P$ be the set of detected paths for each project $P$ from prior stages. The metrics are formally defined as follows:

$$\#VulPath(P) = |\{Path \in Paths^P \mid Path \cap \mathbf{V}_{\text{vul}}^P \neq \emptyset\}|, \quad Rec(P) = \mathbb{1}_{\#VulPath(P)>0},$$
$$\#Detected(\mathcal{D}) = \sum_{P\in\mathcal{D}} Rec(P), \qquad\qquad Prec(P) = \frac{\#VulPath(P)}{|Paths^P|},$$
$$AvgFDR(\mathcal{D}) = \text{avg}_{P\in\mathcal{D},|Paths^P|>0} 1 - Prec(P), \qquad AvgF1(\mathcal{D}) = \frac{1}{|\mathcal{D}|}\sum_{P\in\mathcal{D}} \frac{2\cdot Prec(P)\cdot Rec(P)}{Prec(P)+Rec(P)}.$$

Specifically, a lower *AvgFDR* is preferable, as it indicates a lower ratio of false positives. We note that $Prec(P)$ might sometimes be undefined due to division-by-zero if the detection tool retrieves no path ($|Paths^P| = 0$). Therefore, for *AvgFDR* to be meaningful, we only consider the projects where at least one positive result is produced ($|Paths^P| > 0$). *AvgF1* avoids this issue since $Rec(P) = 0$ when no positive labels exist, forcing the F1 term to be zero regardless of $Prec(P)$.

## 4 CWE-Bench-Java: A Dataset of Security Vulnerabilities in Java

To evaluate our approach, we require a dataset of vulnerable versions of Java projects with several important characteristics: 1) Each benchmark should have relevant **vulnerability metadata**, such as the CWE ID, CVE ID, fix commit, and vulnerable project version, 2) each project in the dataset must be **compilable**, which is a key requirement for static analysis and data flow graph extraction, 3) the projects must be **real-world**, which are typically more complex and hence challenging to analyze compared to synthetic benchmarks, and 4) finally, each vulnerability and its location (e.g., method) in the project must be **validated** so that this information can be used for robust evaluation of vulnerability detection tools. Unfortunately, no existing dataset satisfies all these requirements.

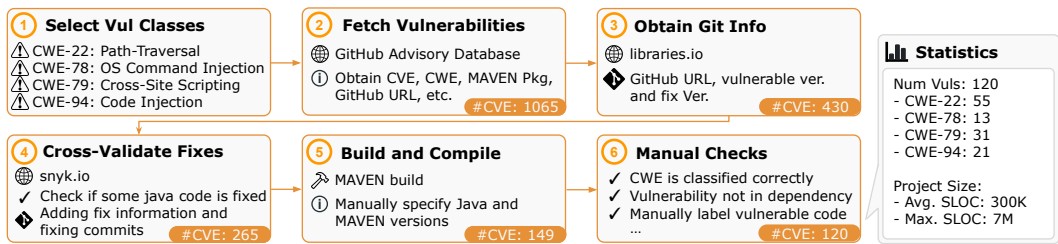

Figure 5: Steps for curating CWE-Bench-Java, and dataset statistics.

To address these requirements, we curate our own dataset of vulnerabilities. For this paper, we focus only on vulnerabilities in Java libraries that are available via the widely used Maven package manager. We choose Java because it is commonly used to develop server-side, Android, and web applications, which are prone to security risks. Further, due to Java's long history, there are many existing CVEs in numerous Java projects that are available for analysis. We initially use the GitHub Advisory database (GitHub, 2024b;c) to obtain such vulnerabilities, and further filter it with cross-validated information from multiple sources, including manual verification. Fig. 5 illustrates the complete set of steps for curating CWE-Bench-Java.

As shown in the statistics (Fig. 5), the sheer size of these projects make them challenging to analyze for any static analysis tool or ML-based tool. Each project in CWE-Bench-Java comes with GitHub information, vulnerable and fix version, CVE metadata, a script that automatically fetches and builds, and the set of program locations that involve the vulnerability.

## 5 EVALUATION

We perform extensive experimental evaluations of IRIS and demonstrate its practical effectiveness in detecting vulnerabilities in real-world Java repositories in CWE-Bench-Java. While additional results and analyses are provided in the Appendix, we address the following key research questions:

- **RQ 1:** How many previously known vulnerabilities can IRIS detect?
- **RQ 2:** Does IRIS detect new, previously unknown vulnerabilities?
- **RQ 3:** How good are the inferred source/sink specifications by IRIS?
- **RQ 4:** How effective are the individual components of IRIS?

### 5.1 EXPERIMENTAL SETUP

We select two closed-source LLMs from OpenAI: GPT-4 (`gpt-4-0125-preview`) and GPT-3.5 (`gpt-3.5-turbo-0125`) for our evaluation. We also select instruction-tuned versions of four open-source LLMs via huggingface API: Llama 3 (L3) 8B and 70B, Qwen-2.5-Coder (Q2.5C) 32B, Gemma-2 (G2) 27B, and DeepSeekCoder (DSC) 7B. For the CodeQL baseline, we use version 2.15.3 and its built-in `Security` queries specifically designed for each CWE. Other baselines included are Facebook Infer (FB Infer), SpotBugs (Lavazza et al., 2020), and Snyk (Snyk.io). We expand further on the other experimental setups in the Appendix.

### 5.2 RQ1: EFFECTIVENESS OF IRIS ON DETECTING EXISTING VULNERABILITIES

The results in Table 1 highlight IRIS's superior performance compared to CodeQL. Specifically, IRIS, when paired with GPT-4, identifies 55 vulnerabilities—28 more than CodeQL. While GPT-4 shows the highest efficacy, smaller, specialized LLMs like DeepSeekCoder 7B still detect 52 vulnerabilities, suggesting that our approach can effectively leverage smaller-scale models, enhancing accessibility. Notably, this increase in detected vulnerabilities does not compromise precision, as evidenced by IRIS's lower average false discovery rate (FDR) with GPT-4 compared to CodeQL. Moreover, IRIS improves average F1 by 0.1, reflecting a better balance between precision and recall. We note that the reported average FDR is a coarse measure as our metrics may consider a true (but unknown) vulnerability found by IRIS as a false positive. Hence, the reported FDR is an upper bound. To get a better sense of detection accuracy, we manually analyzed 50 random alarms reported by IRIS (using GPT-4) and found that 27/50 alarms exhibit potential attack surfaces, *yielding a more refined estimated false discovery rate of 46%.* Hence, IRIS will likely be more effective in practice.

Table 1: Overall performance comparison of CodeQL vs IRIS on Detection Rate (↑), Average FDR (↓), and Average F1 (↑). We present results of IRIS with LLMs including GPT-4 and GPT-3.5, L3 8B and 70B, Q2.5C 32B, G2 27B, and DSC 7B.

| | Method | #Detected (/120) | Detection Rate (%) | Avg FDR (%) | Avg F1 Score |
|---|---|---|---|---|---|
| | CodeQL | 27 | 22.50 | 90.03 | 0.076 |
| **IRIS +** | GPT-4 | **55** (↑ 28) | **45.83** (↑ 23.33) | **84.82** (↓ 5.21) | **0.177** (↑ 0.101) |
| | GPT-3.5 | 47 (↑ 20) | 39.17 (↑ 16.67) | 90.42 (↑ 0.39) | 0.096 (↑ 0.020) |
| | L3 8B | 41 (↑ 14) | 34.17 (↑ 11.67) | 95.55 (↑ 5.52) | 0.058 (↓ 0.018) |
| | L3 70B | 54 (↑ 27) | 45.00 (↑ 22.50) | 90.96 (↑ 0.93) | 0.113 (↑ 0.037) |
| | Q2.5C 32B | 47 (↑ 20) | 39.17 (↑ 16.67) | 92.38 (↑ 2.35) | 0.097 (↑ 0.021) |
| | G2 27B | 45 (↑ 18) | 37.50 (↑ 15.00) | 91.23 (↑ 1.20) | 0.100 (↑ 0.024) |
| | DSC 7B | 52 (↑ 25) | 43.33 (↑ 20.83) | 95.40 (↑ 5.37) | 0.062 (↓ 0.014) |

Table 2 presents a detailed breakdown of detected vulnerabilities, comparing IRIS against various baselines. With the exception of IRIS using Llama-3 8B, which underperforms in detecting CWE-22 vulnerabilities, IRIS consistently outperforms all other baselines. Notably, CWE-78 (OS Command Injection) remains particularly challenging for all LLMs. Our manual investigation revealed that the vulnerability patterns in CWE-78 are highly intricate, often involving OS command injections via gadget-chains (Cao et al., 2023) or external side effects, such as file writes, which are difficult to track using static analysis. This highlights the inherent limitations of static analysis, as opposed to dynamic approaches—an area that we leave for future work.

## 5.3 RQ2: Previously Unknown Vulnerabilities by IRIS

We applied IRIS with GPT-4 to the latest versions of 30 Java projects. Among the 16 inspected projects where IRIS raised at least one alert, we identified 4 vulnerabilities, including 3 instances of path injection (CWE-22) and one case of code-injection (CWE-94). To ensure that these vulnerabilities were indeed uncovered due to IRIS's integration with LLMs, we verified that CodeQL alone did not detect them. We highlight one such vulnerability in Fig. 8. CodeQL was unable to detect this issue due to a missing source specification, while GPT-4 successfully flagged the API endpoint `restoreFromCheckpoint` as a potential entry point for attack.

## 5.4 RQ3: Quality of LLM-Inferred Taint Specifications

The LLM-inferred taint specifications are fundamental to IRIS's effectiveness. To assess the quality of these specifications, we conducted two experiments. First, we used CodeQL's taint specifications as a benchmark to estimate the recall of both source and sink specifications inferred by LLMs (Fig. 6). However, since CodeQL offers a limited set of specifications, we also needed to assess the quality of inferred specifications outside of its known coverage. To this end, we manually analyzed 960 randomly selected samples of LLM-inferred source and sink labels (30 per combination of CWE and LLM) and estimated the overall precision of the specifications (Fig. 7).

**LLM-inferred sinks can replace CodeQL sinks.** Overall, LLMs demonstrated high recall when tested against CodeQL's sink specifications (Fig. 6), with GPT-4 scoring the highest (87.11%).

Table 2: Per-CWE statistics of number of vulnerabilities detected (*#Detected*) by baselines and IRIS. The compared baselines are CodeQL (QL), Facebook Infer (Infer), Spotbugs (SB), and Snyk. The values in parentheses show the differences of detection by IRIS against CodeQL.

| CWE | #Vuls | Baselines | | | | IRIS with | | | | |
|---|---|---|---|---|---|---|---|---|---|---|
| | | QL | Infer | SB | Snyk | GPT-4 | GPT-3.5 | L3 8B | L3 70B | DSC 7B |
| CWE-22 | 55 | 22 | 0 | 2 | 21 | **31** (↑ 9) | 25 (↑ 3) | 19 (↓ 3) | 29 (↑ 7) | 25 (↑ 3) |
| CWE-78 | 13 | 1 | 0 | 1 | 1 | **3** (↑ 2) | 1 (= 0) | 2 (↑ 1) | 2 (↑ 1) | **3** (↑ 2) |
| CWE-79 | 31 | 4 | 0 | 1 | 1 | 13 (↑ 9) | 13 (↑ 9) | 9 (↑ 9) | **14** (↑ 10) | **14** (↑ 10) |
| CWE-94 | 21 | 0 | 0 | 0 | 0 | 8 (↑ 8) | 8 (↑ 8) | **11** (↑ 11) | 9 (↑ 9) | 10 (↑ 10) |
| **All** | 120 | 27 | 0 | 4 | 23 | **55** (↑ 28) | 47 (↑ 20) | 41 (↑ 14) | 54 (↑ 27) | 52 (↑ 25) |

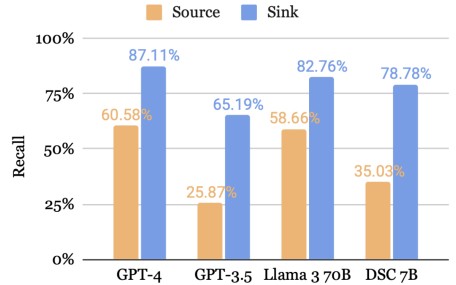

Figure 6: Recall of LLM-inferred taint specifications against CodeQL's taint specifications.

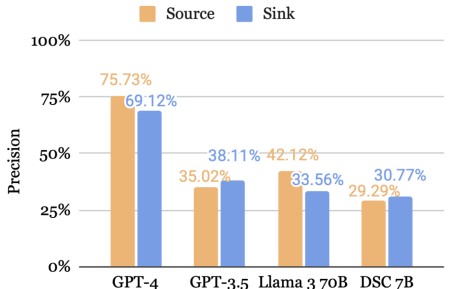

Figure 7: Estimated precision of LLM-inferred specifications on randomly sampled labels.

```
alluxio/dora/core/.../rocks/RocksStore.java

void restoreFromCheckpoint(CheckpointInputStream input) ... {  (1) Source
  // ...
  try (FileOutputStream (2) os = new FileOutputStream(tmpPath)) {
    IOUtils.copy(input, fos);
  }                                                       (3)
  ZipUtils.decompress(Paths.get(mDbPath), tmpZipFilePath, ...);
  // ...
```

```
alluxio/dora/core/.../util/compression/ParallelZipUtils.java

void unzipEntry((6) ile zipFile, ZipArchiveE (5) entry, ...) ... {   (4)
  File outputFile = new File(dirPath.toFile(), entry.getName());
  // ...
  if (!entry.isDirectory()) {                              (7) Sink
    try (FileOutputStream out = new FileOutputStream(outputFile)) {
      // ...
    } // ...
```

Figure 8: A previously unknown vulnerability found in alluxio 2.9.4. The snippets are slightly modified for presentation purpose. A user with database restoration permission may supply a database checkpoint Zip file with malicious entry name. When unzipped, the entry may be written to an arbitrary directory, causing a Zip-Slip vulnerability (CWE-022) that could corrupt the hosting server.

While the recall for source specifications was generally lower, we found that CodeQL tends to over-approximate its source specifications to compensate for a low detection rate. On the other hand, GPT-4 achieved high precision (over 70%) in manual evaluations (Fig. 7), aligning with the lower false discovery rate previously reported in Table 1. For other LLMs, the combination of high recall but lower precision suggests a tendency to over-approximate sink specifications.

**Over-approximating specifications can benefit IRIS.** Although the precision for LLMs other than GPT-4 is lower, over-approximation can actually help address a core limitation of CodeQL—its restricted set of taint specifications. By over-approximating, LLMs expand the coverage of taint analysis, offering a partial solution to CodeQL's limited scope. The impact of this imprecision can be mitigated through contextual analysis as we show next in the ablation studies.

## 5.5 RQ4: ABLATION STUDIES

**Both LLM-inferred sources and sinks are necessary.** Table 3 presents additional results when using either only the source or sink specification from an LLM in IRIS. For this experiment, we only use the results with GPT-4 for comparison. Each row present the number of detected vulnerabilities per CWE. We observe that omitting either source or sink specifications inferred by GPT-4 causes a drastic reduction in overall recall.

**Performance gain of contextual analysis depends on LLM's reasoning capability.** As shown in Fig. 9, contextual analysis is highly necessary for the precision and F1 score improvements. However, only GPT-4, GPT-3.5, and Llama-3 70B see a positive impact after contextual analysis, while the smaller models see negative. The false positive reduction of contextual analysis is the most effective when the LLM possesses decent reasoning capability. Indeed, smaller models are more likely to respond with "vulnerable" than larger models.

## 6 RELATED WORK

**Learning-based approaches for vulnerability detection.** Numerous prior techniques incorporate deep learning for detecting vulnerabilities. This includes Graph Neural Network based models such as Zhou et al. (2019); Chakraborty et al. (2020); Dinella et al. (2020); Hin et al. (2022); Li et al.

Table 3: Ablation on LLM inferred source and sink specifications (CodeQL (QL) versus GPT-4), evaluated using the *#Detected* metrics. When replacing either source or sink with CodeQL specs, we see significantly less vulnerabilities detected.

| CWE | 22 | 78 | 79 | 94 | Total |
|-----|----|----|----|----|-------|
| $Src_{QL} + Snk_{QL}$ | 22 | 1 | 4 | 0 | 27 (↓ 28) |
| $Src_{GPT4} + Snk_{QL}$ | 28 | 3 | 5 | 0 | 36 (↓ 19) |
| $Src_{QL} + Snk_{GPT4}$ | 10 | 1 | 9 | 4 | 24 (↓ 31) |
| $Src_{GPT4} + Snk_{GPT4}$ | 31 | 3 | 13 | 8 | 55 |

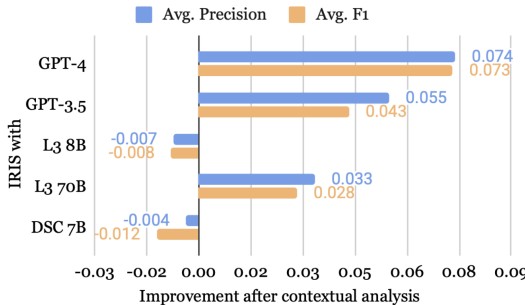

Figure 9: Improvements of Avg. Precision and Avg. F1 after contextual analysis.

(2021a); LSTM-based models such as Li et al. (2020; 2021b); and fine-tuning of Transformer-based models such as Fu & Tantithamthavorn (2022); Steenhoek et al. (2023); Cheng et al. (2022). These approaches focus on method-level detection of vulnerabilities and provide only a binary label classifying a method as vulnerable or not. In contrast, IRIS performs whole-project analysis and provides a distinct code path from a source to a sink and can be tailored for detecting different CWEs. More recently, multiple studies demonstrated that LLMs are not effective at detecting vulnerabilities in real-world code (Steenhoek et al., 2024; Ding et al., 2024; Khare et al., 2023). While these studies only focused on method-level vulnerability detection, it reinforces our motivation that detecting vulnerabilities requires whole-project reasoning, which LLMs currently cannot do alone.

**Static analysis tools.** Apart from CodeQL (Avgustinov et al., 2016), other static analysis tools (CPPCheck; Semgrep, 2023; FlawFinder; FB Infer; Code Checker) also include analyses for vulnerability detection. More general query engines (Scholz et al., 2016; Li et al., 2023b) have also been applied to find program bugs. But these tools are not as feature-rich and effective as CodeQL (Li et al., 2023a; Lipp et al., 2022). Recently, proprietary tools such as Snyk (Snyk.io) and SonarQube (SonarQube) are also gaining in popularity, although sharing the same fundamental limitations of missing specifications and false positives, which IRIS improves upon. We envision our technique to benefit all such tools. Works such as MERLIN (Livshits et al., 2009), Seldon (Chibotaru et al., 2019) and InspectJS (Dutta et al., 2022) tackle the problem of specification inference through probabilistic modeling. Specifically, like IRIS, InspectJS also augments CodeQL with specifications inferred using machine learning. However, InspectJS relies on the quality of seed specifications and requires expensive analysis of each third-party library, which IRIS does not—making it more scalable. Future work could explore incorporating probability estimates for specifications.

**LLM-based approaches for software engineering.** Researchers are increasingly combining LLMs with program reasoning tools for challenging tasks such as fuzzing (Lemieux et al., 2023; Xia et al., 2024), program repair (Xia et al., 2023; Joshi et al., 2023; Xia & Zhang, 2022), and fault localization (Yang et al., 2023). While we are on a similar direction as (Li et al., 2024; Wang et al., 2024), to our knowledge, our work is among the first to combine LLMs with static analysis to detect application-level security vulnerabilities via whole-project analysis.

## 7 CONCLUSION AND LIMITATIONS

We presented IRIS, a novel neuro-symbolic approach that combines LLMs with static analysis for vulnerability detection. We curate a dataset, CWE-Bench-Java, containing 120 security vulnerabilities across four classes in real-world projects. Our results show that systematically combining LLMs with static analysis significantly improves upon traditional static analysis alone in terms of both detected bugs and the alleviation of developer burden.

**Limitations.** There are still many vulnerabilities that IRIS cannot detect. Future approaches may explore a tighter integration of these two tools to improve performance. In addition, IRIS makes numerous calls to LLMs for specification inference and filtering false positives, increasing the potential cost of analysis. While our results on Java are promising, it is unknown if IRIS will perform well on other languages. Moreover, there is still a gap between the IRIS generated report and the report that the developers would like to see. We plan to explore this further in future work.

ACKNOWLEDGMENTS

We thank the anonymous reviewers for their valuable feedback and suggestions that helped improve this work. We also thank Claire Wang for helping with open-sourcing the project. This research was supported by NSF award CCF 2313010.

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

# A  IMPLEMENTATION DETAILS OF IRIS

## A.1  SELECTING CANDIDATE SPECIFICATIONS

While extracting external APIs, we filter out commonly-used Java libraries that are unlikely to contain any potential sources or sinks. Such libraries include testing libraries like JUnit and Hamcrest or mocking libraries like Mockito. While we filter out methods that are defined in the project, we specifically allow methods that are inherited from an external class or interface. An example is the `getResource` method of the generic class `Class` in `java.lang` package, which takes a path as a string and accesses a file in the module. Many projects commonly inherit this class and use this method. If the input path is unchecked, it may lead to a Path-Traversal vulnerability if the path accesses resources outside the given module. Hence, detecting such API usages is crucial.

Taint sources are typically values returned by methods that obtain inputs from external sources, such as response of an HTTP request or a command line argument. Hence, we select external APIs that have a "non-void" return type as candidate sources. Another type of taint sources are commonly seen in Java libraries. When used by downstream libraries, tainted information maybe passed into the library through function calls. Therefore, we also collect the formal parameters for public internal function as source candidates. Due to the excessive amount of such candidates, we pose a further constraint that the public internal function must be directly invoked by a unit test case within the same repository. Here, the test cases are identified by checking whether the residing file path has `src/test` within it.

On the other hand, taint sinks are typically arguments to an external API. This involves *explicit* arguments, such as the `command` argument passed to `Runtime.exec(String command)` method, and *implicit* `this` argument to non-static functions, such as the `file` variable in the function call `file.delete()`. This is the only type of sink that we consider within IRIS.

We note that this is not the entire story as there might be other kinds of sources and sinks. Other types of source candidates include the formal parameter of protected but overridden internal functions (the `req` parameter in `protected HTTPServeletResponse doGet(HTTPServeletRequest req)`), arguments to an impure external function (the `buffer` argument to `void read(byte[] buffer, int size)`), etc. Sink candidates include the return value of public facing functions, thrown exceptions, and even static methods without any parameter (`System.exit()`). Due to the complexity, we do not tackle such kind of sources of sinks in this work. However, we plan to explore further in future work.

## A.2  LLM PROMPTS FOR SPECIFICATION INFERENCE

There are two prompts that we use to query LLM for specification inference. The first one is used to label external APIs as either sources or sinks, illustrated in Listing 1. At a high level, this is a classification task that classifies each API into one of {*Source*, *Sink*, *Taint-Propagator*, *None*}. As shown in the listing, the system prompt involves general instruction about the task and the expected output format, which is JSON. In the user prompt, we give the description of CWE, since the source and sink specifications of external APIs are dependent on the CWE. We additionally give few-shot examples that cover both sources and sinks for the given CWE. At the end, we list out a batch of methods akin to the format of CSV. Notably for sink specifications, we expect the LLM to give extra information about which exact argument to be considered as the sink. This include explicit arguments as well as the implicit `this` argument. We also note that while taint-propagators are included in the prompt, we do not actually use it in the subsequent stages of IRIS. Primarily, the notion of taint-propagator is to help LLMs differentiate between sinks and summary models, which are sometimes mistaken as sinks. In general, we find the prompt to serve the purpose well.

The second prompt, depicted in Listing 2, is used to label the formal parameters of internal APIs as sources. Since we are analyzing internal API, the information such as project `README` and function documentations are commonly available. The goal is to find whether this internal API might be invoked by a downstream library with a malicious input passed to this formal parameter. This information is not CWE specific, hence no CWE information is included in this prompt.

We hypothesize that since LLMs are pre-trained on internet-scale data, they have knowledge about the behavior of widely used libraries and their APIs. Hence, it is natural to ask whether LLMs can be

```
1  System: You are a security expert. You are given a list of APIs to be
       labeled as potential taint sources, sinks, or APIs that propagate
       taints. Taint sources are values that an attacker can use for
       unauthorized and malicious operations when interacting with the
       system. Taint source APIs usually return strings or custom object
       types. Setter methods are typically NOT taint sources. Taint sinks
       are program points that can use tainted data in an unsafe way, which
       directly exposes vulnerability under attack. Taint propagators carry
       tainted information from input to the output without sanitization,
       and typically have non-primitive input and outputs. Return the result
        as a json list with each object in the format:
2
3  { "package": <package name>,
4    "class": <class name>,
5    "method": <method name>,
6    "signature": <signature of the method>,
7    "sink_args": <list of arguments or `this`; empty if the API is not sink
       >,
8    "type": <"source", "sink", or "taint-propagator"> }
9
10 DO NOT OUTPUT ANYTHING OTHER THAN JSON.
11
12
13 User: [CWE_LONG_DESCRIPTION]
14
15 Some example source/sink/taint-propagator methods are:
16 [CWE_SOURCE_SINK_EXAMPLES]
17
18 Among the following methods, \
19 assuming that the arguments passed to the given function is malicious, \
20 what are the functions that are potential source, sink, or taint-
       propagators to [CWE_TITLE] attack (CWE-[CWE_ID])?
21
22 Package,Class,Method,Signature
23 [Package1],[Class1],[Method1],[Signature1]
24 [Package2],[Class2],[Method2],[Signature2]
25 [...]
```

Listing 1: LLM prompt for labelling external APIs as sources or sinks.

used to identify APIs that are relevant as sources or sinks for any vulnerability class. If successful, LLMs can alleviate manual effort, and drastically improve the effectiveness of static analysis tools.

### A.3 DESIGN DECISIONS OF LLM PROMPTS

Here are some reasoning behind the design decisions for the LLM prompts:

1. **Few-shot for labeling external API**: Since the LLM is tasked with labeling source, sink, and taint-propagator APIs, we provide one example for each category, along with a negative example of an API that does not fall into any of these categories. We typically select examples from the Java standard libraries because they are widely used and their labels are readily available.

2. **Zero-shot for labeling internal API**: As labels for internal APIs are not available, we rely on the zero-shot capabilities of the language model. To mitigate potential performance loss, we include additional information, such as documentation associated with important internal APIs.

3. $\pm 5$ **lines surrounding the source and sink location during contextual analysis**: We chose $\pm 5$ lines as a balanced approach to provide sufficient context while managing performance and cost. While technically possible to use a larger window, we observed that excessive context can overwhelm the language model, leading to reduced accuracy. Addi-

```
1  System: You are a security expert. You are given a list of APIs
       implemented in established Java libraries, and you need to identify
       whether some of these APIs could be potentially invoked by downstream
        libraries with malicious end-user (not programmer) inputs. For
       instance, functions that deserialize or parse inputs might be used by
        downstream libraries and would need to add sanitization for
       malicious user inputs. On the other hand, functions like HTTP request
        handlers are typically final and won't be called by a downstream
       package. Utility functions that are not related to the primary
       purpose of the package should also be ignored. Return the result as a
        json list with each object in the format:
2
3  { "package": <package name>,
4    "class": <class name>,
5    "method": <method name>,
6    "signature": <signature>,
7    "tainted_input": <a list of argument names that are potentially tainted
       > }
8
9  In the result list, only keep the functions that might be used by
       downstream libraries and is potentially invoked with malicious end-
       user inputs. Do not output anything other than JSON.
10
11
12 User: You are analyzing the Java package [PROJECT_AUTHOR]/[PROJECT_NAME].
        Here is the package summary:
13
14 [PROJECT_README_SUMMARY]
15
16 Please look at the following public methods in the library and their
       documentations (if present). What are the most important functions
       that look like can be invoked by a downstream Java package that is
       dependent on [PROJECT_NAME], and that the function can be called with
        potentially malicious end-user inputs? If the package does not seem
       to be a library, just return empty list as the result. Utility
       functions that are not related to the primary purpose of the package
       should also be ignored.
17
18 Package,Class,Method,Doc
19 [Package1],[Class1],[Method1],[Documentation1]
20 [Package2],[Class2],[Method2],[Documentation2]
21 [...]
```

Listing 2: LLM prompt for labeling formal parameters of internal APIs as sources.

tionally, a larger context increases computational costs significantly, particularly given the large number of candidate APIs and paths that must be queried.

4. **Selecting subset of nodes in the alarm path during contextual analysis**: We use a hyperparameter $S$ to control the number of intermediate steps included in the prompt. For paths with more than $S$ intermediate steps, we divide the path into $S$ equal segments and select one step from each. This selection prioritizes function calls, as they may indicate sanitizations. If no function call is present, a node is randomly selected from the segment. In our experiments, we observe that setting $S$ to 10 provides a good balance between the cost and accuracy so that the prompt contains enough context and would not be too long.

## A.4 CODEQL QUERIES FOR STATIC ANALYSIS

Listing 3 presents our CodeQL query for Path-Traversal vulnerabilities (CWE 22). Lines 10-29 describe a taint analysis configuration that describes which nodes in the data flow graph should be considered as sources or sinks. Here, Line 12 specifies our custom predicate isLLMDetectedSource that checks whether the method called is taint source based on our

```
1  import java
2  // other imports ...
3  import MySources
4  import MySinks
5
6  /**
7   * A taint-tracking configuration for tracking flow from remote sources
        to the
8   * creation of a path.
9   */
10 module MyTaintedPathConfig implements DataFlow::ConfigSig {
11   predicate isSource(DataFlow::Node source) {
12     isLLMDetectedSource(source)
13   }
14
15   predicate isSink(DataFlow::Node sink) {
16     isLLMDetectedSink(sink)
17   }
18
19   predicate isBarrier(DataFlow::Node sanitizer) {
20     sanitizer.getType() instanceof BoxedType or
21     sanitizer.getType() instanceof PrimitiveType or
22     sanitizer.getType() instanceof NumberType or
23     sanitizer instanceof PathInjectionSanitizer
24   }
25
26   predicate isAdditionalFlowStep(DataFlow::Node n1, DataFlow::Node n2) {
27     isLLMDetectedStep(n1, n2)
28   }
29 }
30
31 /** Tracks flow from remote sources to the creation of a path. */
32 module MyTaintedPathFlow = TaintTracking::Global<MyTaintedPathConfig>;
33
34 from MyTaintedPathFlow::PathNode source, MyTaintedPathFlow::PathNode sink
35 where MyTaintedPathFlow::flowPath(source, sink)
36 select
37   getReportingNode(sink.getNode()),
38   source,
39   sink,
40   "This path depends on a $@.",
41   source.getNode(),
42   sourceType(source.getNode())
```

Listing 3: CodeQL script for detecting vulnerabilities for Path-Traversal (CWE 22).

specifications. Similarly, our predicates `isLLMDetectedSink` checks whether the node is a taint sink based on our specifications. Line 16 checks if a method call or method argument node is a taint sink based on our specifications. We generate the source and sink specifications as predicates in QL file as shown in Listings 4 and 5 respectively. Given a taint configuration and the source and sink specifications, CodeQL can automatically perform taint analysis on a given project.

We use templates to convert LLM inferred specifications into CodeQL queries. There are three kinds of queries:

1. a formal parameter of an internal function as a source;

2. the return value of an external function as a source; and

3. an argument to an external function as a sink.

Example queries for the two kinds of sources are specified in Listing 4, while the example query for the sink is illustrated in Listing 5. As shown in the listings, we not only match on function package,

class, and name, but also match on individual arguments or parameters. Moreover, our query handles generic functions or function in generic classes through the `getSourceDeclaration()` predicate provided by CodeQL. Notably, when the number of inferred specifications is too large, we will split the single predicate into multiple hierarchical ones, improving the CodeQL performance.

```
predicate isLLMDetectedSource(DataFlow::Node src) {
    // Sources: Return value from external APIs
    (
        src.asExpr().(Call).getCallee().getName() = "getName" and
        src.asExpr().(Call).getCallee().getDeclaringType().
    getSourceDeclaration().hasQualifiedName("java.util.zip", "ZipEntry")
    )
    ...
    or
    // Sources: Function formal parameters of internal API
    exists(Parameter p |
        src.asParameter() = p and
        p.getCallable().getName() = "setUserName" and
        p.getCallable().getDeclaringType().getSourceDeclaration().
    hasQualifiedName("org.apache.dolphinscheduler.dao.entity", "DqRule")
    and
        ( p.getName() = "userName" )
    )
    ...
}
```

Listing 4: CodeQL predicate for source specifications.

```
predicate isLLMDetectedSink(DataFlow::Node snk) {
    exists(Call c |
        c.getCallee().getName() = "createTempFile" and
        c.getCallee().getDeclaringType().getSourceDeclaration().
    hasQualifiedName("java.io", "File") and
        ( c.getArgument(0) = snk.asExpr().(Argument) )
    )
    or
    ...
}
```

Listing 5: CodeQL predicate for sink specifications.

## A.5 Visualization of Metrics

We provide a visualization of our *VulDetected* metric in Fig. 10. For evaluation, we assume that the label for a project $P$ is provided as a set of crucial program points $\mathbf{V}_{vul}^P = \{V_1, \ldots, V_n\}$ where the vulnerable paths should pass through. In practice, these are typically the patched methods that can be collected from each vulnerability report. As illustrated in Fig. 10, if at least one detected vulnerable path passes through a fixed location for the given vulnerability, then we consider the vulnerability detected. Let *Paths*$^P$ be the set of detected paths for each project $P$ from prior stages. The vulnerable paths inside project $P$ is given by:

$$VulPaths(P) = \{Path \in Paths^P \mid Path \cap \mathbf{V}_{vul}^P \neq \emptyset\}$$

# B Additional Details of CWE-Bench-Java

## B.1 Details of Dataset Extraction Process

Because we use CodeQL for static analysis, we further need to build each project for CodeQL to extract data flow graphs from the projects. To build each project, we need to determine the correct

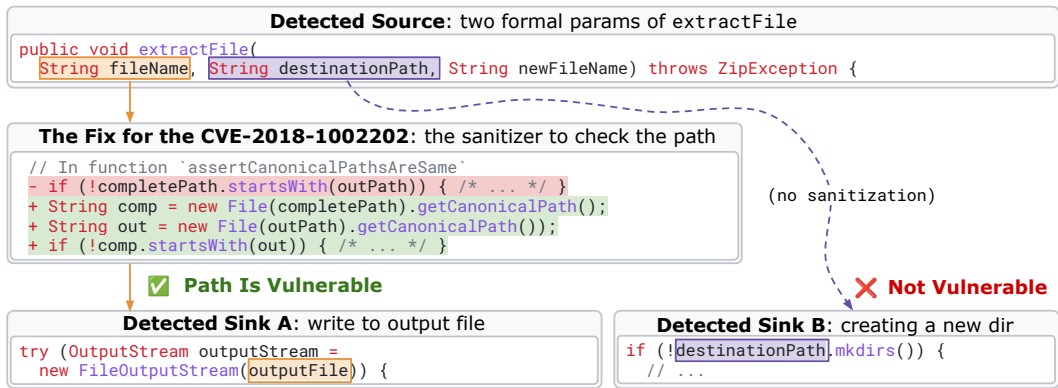

Figure 10: A visualization of our metrics used for vulnerability detection, where the snippets are adapted from Zip4j v2.11.5-2, with slight changes for clearer presentation. While both sinks are potential causes of Path-Traversal (CWE-22), only the dataflow path on the left passes through the fixed sanitizer function. Therefore, we consider only the path on the left as a vulnerability.

Table 4: Vulnerability dataset collection statistics.

| Step | CWE-22 | CWE-78 | CWE-79 | CWE-94 | Total |
|------|--------|--------|--------|--------|-------|
| Initial CVEs | 236 | 39 | 681 | 109 | 1065 |
| W/ Github URL and Version | 119 | 37 | 219 | 55 | 430 |
| W/ Fix Commit | 89 | 27 | 99 | 50 | 265 |
| Compilable | 56 | 17 | 50 | 26 | 149 |
| Fixes in Java Code | 56 | 16 | 25 | 47 | 144 |
| Manual Validation | 55 | 13 | 31 | 21 | 120 |

Java and Maven compiler versions. We developed a semi-automated script that tries to build each project with different combinations of Java and Maven versions. The fourth row in Table 4 presents the number of projects we were able to build successfully. Overall, this results in (⋆) 149 projects.

Finally, we manually check each fix commit and validate whether the commit actually contains a fix to the given CVE in a Java file. For instance, we found that in some cases the fix is in files written in other languages (such as Scala or JSP). While code written in other languages may flow to the Java components in the project during runtime or via compilation, it is not possible to correctly determine if static analysis can correctly detect such a vulnerability. Hence, we exclude such CVEs. Further, we exclude cases where the vulnerability was in a dependency and the fix was just a version upgrade or if the vulnerability was mis-classified. Finally, we end up with (⋆) 120 projects that we evaluate with IRIS. For this task, we divide the CVEs among two co-authors of the project, who independently validate each case. The co-authors cross-check each other's results and discuss together to come up with the final list of projects.

The closest dataset to ours, in terms of features, is the Java dataset curated by Li et al. (2023a), containing 165 CVEs. While we initially considered using their dataset for our work, we found several issues. First, their dataset does not come with build scripts, which makes it hard to automatically run each project with CodeQL. Second, their dataset only has few CVEs for all but one CWE, which makes it difficult to thoroughly analyze a tool for different vulnerability classes. Finally, they do not provide any automated scripts to curate more CVEs. Hence, we curated our own dataset and our framework also allows to easily extend to more vulnerability classes.

## B.2 COMPARISON OF OUR CWE-BENCH-JAVA WITH EXISTING VULNERABILITY DATASETS

We compare CWE-Bench-Java with existing datasets for vulnerability detection in Java, C, and C++ codebases, on the following criteria:

Table 5: Comparison of CWE-Bench-Java with existing vulnerability datasets.

| Dataset | Languages | CVE Info | Real-World | Fix Locations | Compilable | Vetted |
|---|---|---|---|---|---|---|
| BigVul | C/C++ | ✓ | ✓ | ✗ | ✗ | ✗ |
| Reveal | C/C++ | ✗ | ✓ | ✗ | ✗ | ✗ |
| CVEFixes | C/C++, Java, ... | ✓ | ✓ | ✓ | ✓ | ✗ |
| DiverseVul | C/C++ | ✗ | ✓ | ✓ | ✗ | ✗ |
| DeepVD | C/C++ | ✗ | ✓ | ✗ | ✗ | ✗ |
| Juliet | C++, Java | ✗ | ✗ | ✓ | ✓ | ✓ |
| Li et al. (2023a) | Java | ✓ | ✓ | ✓ | ✗ | ✓ |
| SVEN (He & Vechev, 2023) | C++ | ✗ | ✓ | ✓ | ✗ | ✓ |
| CWE-Bench-Java (Ours) | Java | ✓ | ✓ | ✓ | ✓ | ✓ |

1. **CVE Info**: whether CVE Metadata is contained in the dataset;

2. **Real-World**: whether the dataset contains real-world projects;

3. **Fix Locations**: whether the dataset contains fix information about the vulnerabilities;

4. **Compilable**: whether the dataset ensures that the projects are end-to-end and automatically compilable; and

5. **Vetted**: whether the vulnerability in the dataset is manually verified and confirmed.

As shown in Table 5, compared to existing datsets, CWE-Bench-Java, is the only one that checks every criterion. This underscores the significance of our new dataset.

## C EVALUATION DETAILS

### C.1 EXPERIMENTAL SETTINGS

We select two closed-source LLMs from OpenAI: GPT 4 (`gpt-4-0125-preview`) and GPT 3.5 (`gpt-3.5-turbo-0125`) for our evaluation. GPT 4 and GPT 3.5 queries used in the paper are performed through OpenAI API during April and May of 2024.

We also select instruction-tuned versions of six state-of-the-art open-source LLMs via huggingface API: Llama 3 8B and 70B, DeepSeekCoder 7B, QWEN-2.5-Coder 32B, and Gemma-2 27B. To run the open-source LLMs we use two groups of machines: a 2.50GHz Intel Xeon machine, with 40 CPUs, four GeForce RTX 2080 Ti GPUs, and 750GB RAM, and another 3.00GHz Intel Xeon machine with 48 CPUs, 8 A100s, and 1.5T RAM.

We use CodeQL version 2.15.3 as the backbone of our static analysis. We have patched CodeQL with an additional feature that augments the Dataflow edge between throw statement and its closest surrounding try-catch block. We use this CodeQL pull request as the base of our patch.

### C.2 CODEQL BASELINE

For baseline comparison with CodeQL, we use the built-in `Security` queries specifically designed for each CWE that comes with CodeQL 2.15.3. Note that there are multiple security queries for each CWE, and each produce alarms of different levels (error, warning, and recommendation). For each CWE, we take the union of alerts generated by all queries and do not differentiate between alarms of different levels. For instance, there are 3 queries from CodeQL for detecting CWE-22 vulnerabilities, namely `TaintedPath`, `TaintedPathLocal`, and `ZipSlip`. While `TaintedPath` and `ZipSlip` produce error level alarms, `TaintedPathLocal` produces only alarm recommendations. To CodeQL's advantage, all alarms are treated equally in our comparisons.

### C.3 HYPER-PARAMETERS AND FEW-SHOT EXAMPLES

During IRIS, we have 2 prompts that are used to label external and internal APIs. Recall that the prompts contain batched APIs. We use batch size of 20 and 30 for internal and external, respectively.

In terms of few-shot examples passed to labeling external APIs, we use 4 examples for CWE-22, 3 examples for CWE-78, 3 examples for CWE-79, and 3 examples for CWE-94. We use a temperature of 0, maximum tokens to 2048, and top-p of 1 for inference with all the LLMs. For GPT 3.5 and GPT 4, we also fix a seed to mitigate randomness as much as possible.

## C.4 DETAILS OF SELECTED LLMS

We include the versions of the 7 selected LLMs in Table 6.

Table 6: Versions and model IDs of the selected LLMs in our evaluation.

| LLM Version and Size | Model ID |
|---|---|
| GPT 4 | `gpt-4-0125-preview` |
| GPT 3.5 | `gpt-3.5-turbo-0125` |
| Llama 3 8B | `meta-llama/Meta-Llama-3-8B-Instruct` |
| Llama 3 70B | `meta-llama/Meta-Llama-3-70B-Instruct` |
| DeepSeekCoder 7B | `deepseek-ai/deepseek-coder-7b-instruct` |
| Qwen-2.5-Coder 32B Instruct | `Qwen/Qwen2.5-Coder-32B-Instruct` |
| Gemma-2 27B | `google/gemma-2-27b-it` |

## C.5 STATISTICS OF UNIQUE AND RECURRING SPECIFICATIONS

Table 7: Unique source and sink specifications across all projects in CWE-Bench-Java.

| CWE | 22 | 78 | 79 | 94 |
|---|---|---|---|---|
| **#Unique Sources** | 1348 | 899 | 598 | 810 |
| **#Unique Sinks** | 1069 | 575 | 514 | 1281 |

Table 8: Recurring source and sink specifications in CWE-Bench-Java.

| CWE | 22 | 78 | 79 | 94 |
|---|---|---|---|---|
| **#Recurring Sources** | 908 | 232 | 1118 | 626 |
| **#Recurring Sinks** | 919 | 201 | 911 | 961 |

**Continuous taint specification inference is necessary.** Our results show that there is a high number of both unique and recurring sources and sinks. Table 7 presents the number of inferred source and sink specifications that occur only in a single project in CWE-Bench-Java, whereas Table 8 presents the specifications that occur in at least two projects. This indicates that even if previously inferred specifications are useful, a significant number of new relevant APIs still remain and need to be labeled for effective vulnerability detection. This observation strongly motivates the design of IRIS that infers these specifications *on-the-fly* for each project via LLMs, instead of relying on a fixed corpus of specifications like CodeQL.

## C.6 STATISTICS OF INFERRED TAINT SPECIFICATIONS

We show the statistics of inferred taint specifications in Table 9. As shown by the percentage, GPT-4 generates smaller set of sources and sinks than smaller-scale LLMs like DeepSeekCoder 7B.

Table 9: Ratio of API candidates labeled as source (S) or sink (N) by GPT-4 and DeepSeekCoder (DSC) 7B, per CWE and in total.

| CWE | #Cand. | GPT-4 | | DSC 7B | |
|---|---|---|---|---|---|
| | | %S | %N | %S | %N |
| 22 | 130,974 | 2.03% | 1.90% | 4.27% | 4.01% |
| 78 | 25,605 | 4.73% | 3.37% | 3.67% | 3.33% |
| 79 | 37,138 | 5.69% | 4.69% | 4.28% | 4.56% |
| 94 | 36,325 | 5.12% | 7.83% | 6.11% | 6.21% |
| **Total** | 230,042 | 3.41% | 3.45% | 4.50% | 4.37% |

## C.7 Error Analysis: Cause of Undetected Vulnerabilities

In general, we find the following three main causes of undetected vulnerabilities by employing IRIS:

1. **Vulnerability cannot be modeled by simple taint dataflows**: for instance, the vulnerability CVE-2020-11977 (CWE-94) is manifested by an unexpected exit(1) call, with no direct taint dataflow going into it. In fact, the taint dataflow goes into the condition of an "if" statement surrounding the exit. In this case, the model of sink API cannot capture the vulnerability.

2. **Missing dataflow edge due to side-effects**: for instance a taint information is written to a temporary file, and is later read from the same file, subsequently causing a vulnerability. However, the dataflow edge is carried through side-effects, which is not captured by CodeQL.

3. **Missing dataflow edge due to unspecified library usage**: The vulnerability can only be manifested through a concrete usage of the library; but within the library itself there is no possible dataflow to connect the source and the sink.

Overall, we view the above as general limitations for the static analysis. In terms of LLM induced false negatives, here are the two main failure modes:

1. **Missing taint-propagator labels from LLM**: this would cause missing dataflow edges stopping source to flow to sink.

2. **Missing source or sink specifications from LLM**: if there is no relevant source/sink specification then the static analysis tool would not have the anchor for analysis.

# D Analysis Runtime

We include the full table containing statistics to provide more details about projects and our analysis (Table 10). For each project, we present its corresponding CWE ID, the lines-of-code (SLOC), the time it takes to run the full analysis, the number candidate APIs and the number of labeled source and sinks by Llama 3 8B. We also color code cells of interest: For SLOC, we mark a cell as red if >1M; yellow if >100k. For Time, we mark a cell as red if ≥1h; yellow if ≥5m. For the number of candidates, we mark a cell as red if >10k. Lastly for the numbers of sources and sinks, we mark a cell as red if the number is larger than 200.

Table 10: Details of analysis runtime, candidates, and inferred sources and sinks for all projects (Llama 3 8B).

| CWE-ID | Project | SLOC | Time | #Candidates | #Sources | #Sinks |
|--------|---------|------|------|-------------|----------|--------|
| 22 | DSpace | 218.2K | 15s | 3.61K | 162 | 217 |
| 22 | spark | 10.7K | 1m | 679 | 35 | 27 |
| 22 | spark | 9.77K | 57s | 598 | 33 | 22 |
| 22 | wildfly | 496.28K | 4m | 14.13K | 457 | 425 |
| 22 | vertx-web | 51.01K | 1m | 2.06K | 80 | 77 |
| 22 | camel | 1.16M | 8m | 293 | 22 | 9 |
| 22 | hutool | 135.34K | 4m | 6.17K | 115 | 211 |
| 22 | tika | 106.3K | 2m | 3.84K | 277 | 177 |
| 22 | retrofit | 19.28K | 1m | 880 | 28 | 13 |
| 22 | jspwiki | 149.45K | 1m | 1.83K | 62 | 80 |
| 22 | camel | 1.21M | 11m | 4.43K | 53 | 80 |
| 22 | tapestry-5 | 160.06K | 1m | 3.04K | 91 | 66 |
| 22 | spring-cloud-co | 18.56K | 1m | 1.16K | 40 | 64 |
| 22 | spring-cloud-co | 18.44K | 59s | 1.16K | 40 | 64 |
| 22 | rocketmq | 94.64K | 1m | 2.78K | 28 | 54 |
| 22 | mpxj | 181.55K | 1m | 1.6K | 37 | 43 |
| 22 | flink | 1.14M | 2h | 5.16K | 39 | 61 |

| | | | | | | |
|---|---|---|---|---|---|---|
| 22 | java | 1M | 2m | 8.04K | 96 | 41 |
| 22 | commons-io | 29.24K | 58s | 1.07K | 12 | 47 |
| 22 | karaf | 135.22K | 1m | 5.43K | 150 | 210 |
| 22 | james-project | 434.32K | 4m | 14.58K | 209 | 226 |
| 22 | vertx-web | 49.28K | 1m | 2.36K | 83 | 96 |
| 22 | esapi-java-lega | 35.26K | 59s | 1.48K | 43 | 67 |
| 22 | xwiki-commons | 103.05K | 1m | 3.76K | 104 | 137 |
| 22 | zip4j | 16.78K | 58s | 532 | 6 | 34 |
| 22 | one-java-agent | 5.19K | 51s | 327 | 11 | 20 |
| 22 | myfaces | 161.02K | 1m | 2.4K | 68 | 44 |
| 22 | undertow | 86.03K | 1m | 2.58K | 66 | 93 |
| 22 | DependencyCheck | 28.57K | 1m | 1.23K | 47 | 66 |
| 22 | plexus-archiver | 13.04K | 51s | 573 | 34 | 47 |
| 22 | plexus-archiver | 13.04K | 51s | 573 | 34 | 47 |
| 22 | zt-zip | 6.64K | 52s | 337 | 14 | 31 |
| 22 | curekit | 511 | 43s | 73 | 2 | 4 |
| 22 | aws-sdk-java | 7.72M | 38m | 12K | 62 | 65 |
| 22 | venice | 115.44K | 1m | 2.27K | 36 | 79 |
| 22 | DSpace | 237.33K | 1m | 3.67K | 179 | 233 |
| 22 | Payara | 1.12M | 7m | 16.05K | 379 | 427 |
| 22 | DSpace | 237.33K | 1m | 3.67K | 179 | 233 |
| 22 | goomph | 12.68K | 59s | 1.12K | 35 | 111 |
| 22 | dolphinschedule | 90.69K | 1m | 3.36K | 65 | 92 |
| 22 | dolphinschedule | 91.94K | 1m | 3.4K | 65 | 92 |
| 22 | testng | 95.53K | 1m | 2.08K | 33 | 73 |
| 22 | uima-uimaj | 226.81K | 2m | 5.66K | 103 | 176 |
| 22 | keycloak | 614.82K | 12m | 13.34K | 325 | 252 |
| 22 | glassfish | 1.19M | 5m | 12.19K | 293 | 346 |
| 22 | graylog2-server | 382K | 4m | 13.3K | 227 | 171 |
| 22 | mina-sshd | 130.14K | 1m | 3.64K | 52 | 120 |
| 22 | shiro | 38.68K | 1m | 1.5K | 41 | 42 |
| 22 | plexus-archiver | 15.51K | 57s | 666 | 37 | 56 |
| 22 | plexus-utils | 23.3K | 58s | 754 | 16 | 36 |
| 22 | yamcs | 693.6K | 2m | 11K | 98 | 113 |
| 22 | yamcs | 693.6K | 2m | 11K | 98 | 113 |
| 22 | shiro | 38.94K | 1m | 1.53K | 41 | 43 |
| 22 | sling-org-apach | 8.34K | 54s | 695 | 28 | 25 |
| 78 | xstream | 43.49K | 1m | 1.39K | 91 | 30 |
| 78 | xstream | 59.79K | 1m | 1.64K | 107 | 42 |
| 78 | xstream | 52.25K | 1m | 1.64K | 107 | 43 |
| 78 | docker-commons- | 2.79K | 54s | 362 | 25 | 20 |
| 78 | workflow-cps-pl | 17.02K | 1m | 1.38K | 72 | 61 |
| 78 | workflow-cps-gl | 4.31K | 55s | 523 | 40 | 38 |
| 78 | workflow-multib | 3.45K | 53s | 500 | 30 | 30 |
| 78 | activemq | 442.42K | 4m | 6.34K | 234 | 192 |
| 78 | plexus-utils | 22.76K | 1m | 714 | 34 | 17 |
| 78 | git-client-plug | 16.41K | 1m | 1.06K | 83 | 50 |
| 78 | perfecto-plugin | 667 | 54s | 107 | 5 | 10 |
| 78 | nifi | 915.95K | 11m | 22.44K | 894 | 614 |
| 78 | script-security | 8.17K | 1m | 678 | 40 | 46 |
| 79 | antisamy | 6.38K | 57s | 381 | 42 | 33 |
| 79 | antisamy | 6.38K | 56s | 381 | 42 | 33 |
| 79 | jspwiki | 149.33K | 1m | 1.84K | 156 | 110 |
| 79 | jspwiki | 149.33K | 1m | 1.84K | 156 | 110 |
| 79 | jspwiki | 149.33K | 1m | 1.84K | 156 | 110 |
| 79 | jspwiki | 157.09K | 1m | 1.85K | 157 | 110 |
| 79 | hibernate-valid | 93.6K | 1m | 2.06K | 79 | 57 |
| 79 | cxf | 798.53K | 1h | 16.54K | 821 | 756 |
| 79 | xxl-job | 9.32K | 60s | 540 | 42 | 41 |

| 79 | json-sanitizer | 1.47K | 52s | 67 | 4 | 5 |
|----|----------------|-------|-----|-----|-----|-----|
| 79 | hawkbit | 112.09K | 1m | 4.07K | 144 | 151 |
| 79 | nacos | 203.78K | 2m | 4.08K | 201 | 139 |
| 79 | antisamy | 4.93K | 1m | 362 | 43 | 34 |
| 79 | esapi-java-lega | 35.26K | 1m | 1.48K | 107 | 85 |
| 79 | antisamy | 5.14K | 1m | 377 | 44 | 36 |
| 79 | jolokia | 29.97K | 1m | 1.66K | 117 | 97 |
| 79 | keycloak | 60.6K | 1m | 2.1K | 170 | 136 |
| 79 | cxf | 722.83K | 15m | 15.09K | 766 | 710 |
| 79 | sling-org-apach | 1.37K | 55s | 136 | 4 | 13 |
| 79 | DSpace | 237.33K | 2m | 3.67K | 347 | 320 |
| 79 | keycloak | 615.6K | 3h | 13.37K | 606 | 461 |
| 79 | keycloak | 615.6K | 3h | 13.37K | 606 | 461 |
| 79 | xwiki-commons | 105.92K | 1m | 3.94K | 244 | 151 |
| 79 | xwiki-commons | 105.94K | 1m | 3.94K | 244 | 151 |
| 79 | xwiki-rendering | 97.01K | 1m | 1.22K | 73 | 92 |
| 79 | xwiki-commons | 106.87K | 1m | 3.99K | 254 | 161 |
| 79 | jspwiki | 158.7K | 1m | 2.22K | 176 | 126 |
| 79 | keycloak | 617.15K | 4h | 14.04K | 643 | 479 |
| 79 | xwiki-commons | 107.09K | 1m | 3.03K | 209 | 143 |
| 79 | jstachio | 53.02K | 54s | 792 | 40 | 46 |
| 79 | xwiki-rendering | 97.63K | 1m | 1.24K | 74 | 92 |
| 94 | spring-security | 43.9K | 1m | 1.83K | 120 | 176 |
| 94 | xstream | 52.25K | 1m | 1.64K | 111 | 145 |
| 94 | cron-utils | 13.08K | 1m | 476 | 13 | 26 |
| 94 | struts | 160.51K | 12m | 4.39K | 301 | 357 |
| 94 | activemq | 547.68K | 1h | 7.55K | 370 | 607 |
| 94 | spring-framewor | 666.11K | 45m | 17.71K | 688 | 846 |
| 94 | spring-cloud-ga | 25.56K | 1m | 2.01K | 130 | 153 |
| 94 | dubbo | 175.63K | 2m | 6.73K | 342 | 383 |
| 94 | incubator-dubbo | 96.35K | 1m | 3.68K | 194 | 255 |
| 94 | spring-security | 57.34K | 1m | 2.43K | 192 | 234 |
| 94 | kubernetes-clie | 806.35K | 3m | 2.33K | 93 | 130 |
| 94 | commons-text | 24.87K | 1m | 962 | 40 | 47 |
| 94 | ff4j | 46.21K | 1m | 2.39K | 133 | 274 |
| 94 | spring-boot-adm | 18.29K | 1m | 1.83K | 92 | 157 |
| 94 | sqlite-jdbc | 17.71K | 59s | 732 | 50 | 74 |
| 94 | nifi | 993.76K | 25m | 57 | 2 | 11 |
| 94 | rocketmq | 108.39K | 2m | 3.4K | 117 | 164 |
| 94 | nifi | 1.01M | 27m | 261 | 27 | 24 |
| 94 | rocketmq | 197.78K | 2m | 6.28K | 205 | 252 |
| 94 | dolphinschedule | 154.95K | 4m | 5.78K | 229 | 353 |
| 94 | dolphinschedule | 154.95K | 4m | 5.78K | 229 | 353 |

