# OpenReview forum: "IRIS: LLM-Assisted Static Analysis for Detecting Security Vulnerabilities"
_ICLR.cc/2025/Conference — ICLR 2025 Poster_

### Official Review · Reviewer_nTbW · 2024-10-25

**Soundness:** 3
**Presentation:** 3
**Contribution:** 3
**Rating:** 8
**Confidence:** 4

**Summary:**

The paper claims to be the first work to combine LLMs with static analysis to detect application-level security vulnerability via whole-project analysis. They propose a contextual analysis technique with LLMs that reduces false positive alarms and minimizes the triaging effort for developers. Their insight is that encoding the code context and path-sensitive information in the prompt elicits more reliable reasoning from LLMs. They curated a benchmarking dataset of real-world JAVA programs that contain 120 security vulnerabilities. They evaluate IRIS on the Java dataset using eight diverse open- and closed-source LLMs. Doing an extensive evaluation, they show that their method IRIS obtains the best results by detecting 55 vulnerabilities, which is 28 (103.7%) more than CodeQL, the existing best-performing static analyzer.

**Strengths:**

1. The paper addresses the long-standing security problem of decreasing the false positive rate of vulnerability detection while maintaining high enough accuracy.
2. They have done a thorough evaluation using both open—and close-source LLMs and different static analysis baselines, such as Facebook Infer, SpotBugs, and Snyk.
3. They introduced a new dataset with important characteristics such as containing vulnerability metadata, being compilable, demonstrating real work, and being validated.

**Weaknesses:**

1. Tested only on Java codebase.
2. They have excluded code written in other languages that may flow to the Java components in the project during runtime or via compilation.
3. An evaluation of how many detection misses were due to CodeQL's fault and how many were due to LLM's incorrect specification or filtering fault.

**Questions:**

1. They hypothesize that since LLMs are pre-trained on internet-scale data, they know about the behavior of widely used libraries and their APIs. So, can any new package or Java module not processed by the LLMs cause detection or specification generation issues?
2. I want more details on the specification tuples, such as what are all the possible values for the type of node to match in G and what is the upper bound for N from the N-tuple of F?
3. The authors mentioned including ±5 lines surrounding the exact source and sink location and the enclosing function and class. How did they come up with the "5" value? Can it be arbitrarily large, bounded by the LLM's token limitation?
4. I would like to know if, in the event that CodeQL does not detect a vulnerability even with correctly labeled specifications,will it be marked as negative? Can you give me any statistics on how many such instances there are? If I understand correctly, CodeQL is still a critical part of the vulnerability detection pipeline. LLM just ensures that the data fed into the static analyzer is of quality, and once detection is made, LLM helps weed out false positives?
5. Has it ever happened that LLM made a mistake and marked a positive case as negative even after the positive detection of a vulnerable path by CodeQL? I am trying to understand how accurate the LLM-based filtering method is and how drastic the change in explanation for detection will be based on the different specifications provided.

---

> ### Author Response · Authors · 2024-11-23
>
> > **Q1: “Can any new package or Java module not processed by the LLMs cause detection or specification generation issues?”**
>
> We acknowledge that LLMs may not have prior exposure to external APIs from private projects. However, they can still provide best-effort guesses based on the method information available, which may include JavaDoc documentation. This is a significant improvement over traditional methods, which rely solely on human labels and would be completely ineffective in such scenarios. That said, we acknowledge that there may be better methodologies for generalization to unseen cases, and we aim to explore these in future work.
>
> > **Q2: “what are all the possible values for the type of node to match in G and what is the upper bound for N from the N-tuple of F”**
>
> The type of nodes could be
>
> - `argument[i]`, denoting the i-th argument of an external  function call
> - `argument[this]`, denoting the “this” argument in an external function call (e.g. `parser` in the call `parser.parse(arg1, arg2)`)
> - `parameter[i]`, denoting the i-th parameter of an internal function definition
> - `parameter[this]`, denoting the “this” argument of an internal function definition
> - `return_value`, denoting the return value of an external function call
>
> The upper bound for N corresponds to the maximum number of arguments of a given API, which, in our CWE-Bench-Java, is 32.
>
> > **Q3: “in the event that CodeQL does not detect a vulnerability even with correctly labeled specifications, will it be marked as negative? Can you give me any statistics on how many such instances there are? If I understand correctly, CodeQL is still a critical part of the vulnerability detection pipeline. LLM just ensures that the data fed into the static analyzer is of quality, and once detection is made, LLM helps weed out false positives?”**
>
> If taint specifications are all correct and that the data-flow is capable of modeling the vulnerability, then CodeQL **will** be able to detect a vulnerability. In case that the data-flow cannot model the vulnerability, CodeQL will not be able to detect the vulnerability, and the result **will** be marked as **negative**. Therefore, CodeQL is still a critical part of the vulnerability detection pipeline.
>
> In our dataset, we observe at least 12 such cases where normal taint data-flow cannot model the vulnerability. However, the general statistics relevant to our evaluation is very hard to retrieve. This is due to the missing ground-truth source and sink labels as well as the sheer size of our projects.
>
> > **Q4: “How did they come up with the "±5" value? Can it be arbitrarily large, bounded by the LLM's token limitation?”**
>
> In our experiments, we chose ±5 as a balanced approach to provide sufficient context while maintaining performance and cost. While it is technically possible to use a larger window that encompasses the entire function or even the class definition, we found that too much context can overwhelm the LLM, leading to reduced accuracy. Furthermore, increasing the context size substantially raises computational costs—especially given the large number of candidate APIs and paths that must be queried.
>
> Looking ahead, we plan to explore more targeted methods for context selection. Instead of using a fixed number of surrounding lines, we could incorporate only the most relevant variable definitions, class and function signatures, and other critical elements. However, achieving this balance between including relevant information and maintaining manageable context sizes will require further investigation, which we leave for future work.
>
> > **Q5: “Has it ever happened that LLM made a mistake and marked a positive case as negative even after the positive detection of a vulnerable path by CodeQL?”**
>
> Yes, this occurs occasionally. For example, there was a path traversal vulnerability (CWE-22) caused by an insufficient Regex pattern used to sanitize user input. While CodeQL successfully reported this path, the LLM failed to recognize the vulnerability during contextual analysis. This was because the Regex pattern was defined as a static global variable outside the provided context, making it inaccessible to the LLM. As a result, the LLM incorrectly assumed the input was properly sanitized and flagged the path as non-vulnerable.
>
> As mentioned in our response to Q4, we plan to explore a more targeted method for context retrieval, to improve the precision of contextual analysis.

---

### Official Review · Reviewer_3b2y · 2024-11-03

**Soundness:** 3
**Presentation:** 4
**Contribution:** 3
**Rating:** 6
**Confidence:** 4

**Summary:**

This paper proposes using a neuro-symbolic approach, combining LLM and static taint analysis tool CodeQL, for whole-repository level, Java vulnerability detection. The authors curate CWE-Bench-Java, a vulnerability dataset of Java projects. Evaluation results show that IRIS is able to detect significantly more vulnerabilities compared with the baselines and are able to uncover previously unknown vulnerabilities.

**Strengths:**

- IRIS includes a LLM filtering step via contextual analysis, as the use of LLM for sources and sinks predictions may incur many spurious alerts. The ablation study shows that this step can greatly improve Avg F1 score for larger models.
- IRIS significantly outperforms prior methods in terms of the number of vulnerabilities detected and is able to uncover previously unknown vulnerability.

- IRIS focus on project-level vulnerability detection on CWE-Bench-Java, which is inherently a challenging task as the projects are of large sizes.

**Weaknesses:**

- The false discovery rate of IRIS is very high (~90%) and the F1 score is low (around 0.1). The Avg FDR and Avg F1 scores for IRIS + GPT-3.5 / Lamma-3 / DeepSeekCoder are either worse or comparable to the CodeQL baseline.
  - What are the causes of the undetected vulnerabilities (among the 120)?
- A more rigorous discussion or evaluation should be conducted beyond random sampling of 50 alarms to argue that the actual false discovery rate should be much lower (line 397-399).
- Models other than GPT-4 are over-approximating the specifications (Figure 7). The paper would benefit more from method designs to improve LLM-inferred specifications to lower the false discovery rate to save manual efforts of triaging through the alerts.

- Presentation
  - Line 377: "IRIS's superior performance compared to CodeQL:" Should be more careful with the language here, as Table 1 doesn't show its superiority on Avg FDR and F1 metrics.

**Questions:**

1. Can you add related works where the vulnerability is considered detected when the vulnerable path passes through some crucial program points (line 297-298)?
2. Can you explain the design choices in Section 3? It seems to me that these may affect LLM's taints specification inference and predictions of false positives.
   1. Few-shot (3-shot) for external APIs and zero-shot for internal APIs: does the number of shots affect the precision of LLM-inferred specifications?
   2. $\pm$ 5 lines surrounding the source and sink location (line 261),
   3. A subset of nodes (line 263 - 264): how are they selected?

---

> ### Author Response · Authors · 2024-11-23
>
> > **Q1: “What are the causes of the undetected vulnerabilities (among the 120)?”**
>
> We thank the reviewer for the question, we will include the following error analysis for false negatives into our paper.
>
> - _Vulnerability cannot be modeled by simple taint dataflows_: for instance, the vulnerability CVE-2020-11977 (CWE-94) is manifested by an unexpected `exit(1)` call, with no direct taint dataflow going into it. In fact, the taint dataflow goes into the condition of an “if” statement surrounding the exit. In this case, the model of sink API cannot capture the vulnerability.
>
> - _Missing data-flow edge due to side-effects_: for instance a taint information is written to a temporary file, and is later read from the same file, subsequently causing a vulnerability. However, the dataflow edge is carried through side-effects, which is not captured by CodeQL.
>
> - _Missing data-flow edge due to unspecified library usage_: The vulnerability can only be manifested through a concrete usage of the library; but within the library itself there is no possible dataflow to connect the source and the sink.
>
> In general, we view the above as general limitations for the static analysis. We can hopefully resolve them with query synthesis, and we leave these for future work. In terms of LLM induced false negatives, here are the two main failure modes:
>
> - _Missing taint-propagator labels from LLM_: this would cause missing data-flow edges stopping source to flow to sink.
>
> - _Missing source/sink specifications from LLM_: if there is no relevant source/sink specification then the static analysis tool would not have the anchor for analysis.
>
> > **Q2: Discussion on “vulnerability is considered detected when the vulnerable path passes through some crucial program points”**
>
> According to [1], the function-level metric is not enough, and the calling context surrounding the vulnerability matters. We adopt the strategy based on human understanding of vulnerability, where we look at the path between the places where taint is initialized and manifested. This path-based structural coverage criteria is widely adopted in software testing [2]. In the empirical study [3], the authors look at the evaluation metric which uses the fix of the vulnerability as the crucial program point.
>
> - [1]: Top Score on the Wrong Exam: On Benchmarking in Machine Learning for Vulnerability Detection, Risse et. al., 2024
> - [2]: Introduction to software testing, P Ammann, 2008
> - [3]: How Many of All Bugs Do We Find? A Study of Static Bug Detectors, Habib et. al., ASE 2018
>
> > **Q3: “A more rigorous discussion or evaluation should be conducted beyond random sampling of 50 alarms to argue that the actual false discovery rate should be much lower (line 397-399).”**
>
> We thank the reviewer for raising this important concern. Our manual analysis primarily focuses on evaluating the potential attack surface and the manifestation of vulnerabilities, which aligns with the key factors used to determine severity according to the Common Vulnerability Scoring System (CVSS) [4]. For example, in our analysis, we label an alarm as a vulnerability if it presents a "local" attack vector, even if the resulting CVSS score may be relatively low.
>
> We agree that a more thorough, CVSS-based evaluation of the alarms reported by IRIS would provide additional rigor. However, it is outside the scope of our current study. We acknowledge this as a valuable direction for future work. In our revised version, we will elaborate on the detailed criteria for our manual analysis.
>
> - [4]: Vulnerability Metrics, National Vulnerability Database, https://nvd.nist.gov/vuln-metrics/cvss

---

> > ### Author Response · Authors · 2024-11-23
> >
> > > **Q4: “Can you explain the design choices in Section 3?”**
> >
> > We thank the reviewer for pointing this out. We will add the following descriptions to our paper:
> >
> > **few-shot for external API**: Since the LLM is tasked with labeling source, sink, and taint-propagator APIs, we provide one example for each category, along with a negative example of an API that does not fall into any of these categories. We typically select examples from the Java standard libraries because they are widely used and their labels are readily available.
> >
> > **zero-shot for internal API**: As labels for internal APIs are not available, we rely on the zero-shot capabilities of the language model. To mitigate potential performance loss, we include additional information, such as documentation associated with important internal APIs.
> >
> > **5 lines surrounding the source and sink location**: We chose ±5 lines as a balanced approach to provide sufficient context while managing performance and cost. While technically possible to use a larger window, we observed that excessive context can overwhelm the language model, leading to reduced accuracy. Additionally, a larger context increases computational costs significantly, particularly given the large number of candidate APIs and paths that must be queried.
> >
> > **a subset of nodes are selected**: We use a hyperparameter $S$ to control the number of intermediate steps included in the prompt. For paths with more than $S$ intermediate steps, we divide the path into $S$ equal segments and select one step from each. This selection prioritizes function calls, as they may indicate sanitizations. If no function call is present, a node is randomly selected from the segment. In our experiments, we observe that setting $S$ to 10 provides a good balance between the cost and accuracy so that the prompt contains enough context and would not be too long.
> >
> > > **Q5: “Line 377: ‘IRIS's superior performance compared to CodeQL:’ Should be more careful with the language here, as Table 1 doesn't show its superiority on Avg FDR and F1 metrics.”**
> >
> > We thank the reviewer for pointing out this oversight. We will revise the sentence as follows:
> >
> > "The results in Table 1 compare IRIS and CodeQL, highlighting IRIS's superior performance specifically when paired with GPT-4."

---

> > > ### Author Response · Authors · 2024-11-26
> > >
> > > We appreciate your valuable comments on our paper. We have prepared a rebuttal and tried our best to address your concerns. We are willing to answer any unresolved or further questions that you may have regarding our rebuttal if time is allowed. If our rebuttal has addressed your concerns, we would appreciate it if you would let us know if your final thoughts. Additionally, we will be happy to answer any further questions regarding the paper. Thank you for your time and consideration.

---

> > > > ### Comment · Reviewer_3b2y · 2024-11-26
> > > >
> > > > Thank you for the author's clarification. Overall, it is a great work, and I particularly like the neuro-symbolic approach of combining LLM and static taint analysis tool like CodeQL. It is also great to see that IRIS can uncover previously unknown vulnerabilities.
> > > >
> > > > My main concern is still on the high false discovery rate and low F1 score of IRIS, which don't show much improvement from the CodeQL baseline. There is no strong evidence showing that the actual false discovery rate should be much lower. This may indicate that manual efforts are needed to triage through many potentially false security alerts.
> > > >
> > > > Therefore, I would like to maintain my current score.

---

### Official Review · Reviewer_j2y3 · 2024-11-04

**Soundness:** 2
**Presentation:** 2
**Contribution:** 2
**Rating:** 6
**Confidence:** 4

**Summary:**

The paper proposed IRIS, a neuro-symbolic approach that leverages LLMs to infer taint specifications and perform contextual analysis for security vulnerability detection in combination with static analysis tools like CodeQL. The evaluation comes with a newly curated dataset CWE-Bench-Java, comprising 120 manually validated security vulnerabilities in real-world Java projects. The experiment result shows that IRIS with GPT-4 detects 28 more vulnerabilities than CodeQL alone.

**Strengths:**

1. Originality: The paper proposed a new method that combined LLM with static analysis tool CodeQL to infer taint specifications and perform contextual analysis.
2. Quality: The experiment shows huge improvement on bug detection effectiveness
3. Clarity: The paper is well-written and implementation details like prompts and QL scripts are provided in appendix.
4. Significance: The paper addressed challenges of static taint analysis like false negatives due to missing taint specifications, contributing to the field in the long term.

**Weaknesses:**

1. The paper doesn't address potential data leakage and memorization by LLMs, since they are asked to infer sources/taints purely based on method name and signature instead of implementation. The method may work well for projects where the APIs (sources and taints) are already publicly known, but not for private projects or less well-known projects.
2. The comparison with baseline CodeQL may not be fair enough as the LLM-based method will always have advantages on bug detection numbers due to more source/sinks. It might be more convincing if LLM-based approach can be compared with heuristics shown in prompts like "Taint source APIs usually return strings or custom object types." to see that LLM's advantage against heuristics like name and signature type matching.

**Questions:**

1. As mentioned in the limitation section, "IRIS makes numerous calls to LLMs for specification inference and filtering false positives, increasing the potential cost of analysis" I would like to know the actual cost of LLM call for IRIS on CWE-Bench-Java in total and on average.

---

> ### Author Response · Authors · 2024-11-23
>
> > **Q1: “potential data leakage and memorization by LLMs, since they are asked to infer sources/taints purely based on method name and signature instead of implementation”**
>
> While we acknowledge that our method relies on the LLM’s existing knowledge, we argue that the risk of data leakage is minimal. The labels for source/sink APIs and their specific types are typically not publicly available on the internet. Moreover, the paths summarized by CodeQL are even less familiar to LLMs, making it highly unlikely that these were seen during training. This approach thus represents a genuine challenge for the LLM’s capability in knowledge transfer and logical reasoning.
>
> We would also like to clarify that while LLMs may not have prior exposure to external APIs from private projects, they can still provide best-effort guesses based on the information available. This is a significant improvement over traditional methods, which rely solely on human labels and would be completely ineffective in such scenarios. That said, we acknowledge that there may be better methodologies for generalization to unseen cases, and we aim to explore these in future work.
>
> Regarding the sole reliance on method names and signatures, we emphasize that this was a deliberate design decision made after carefully balancing accuracy and cost. While including full function implementations might improve accuracy, the vast number of candidates requiring labeling (Table 11) makes it impractical to query LLMs for specification inference with complete implementation details. Further, analyzing the entire function implementation is itself a challenging task for LLMs.
>
> > **Q2: “The comparison with baseline CodeQL may not be fair enough as the LLM-based method will always have advantages due to more source/sinks. … LLM's advantage against heuristics like name and signature type matching.”**
>
> We thank the reviewer for raising this insightful question. While it is true that LLMs can propose more source/sink candidates, two critical factors must be considered: (1) the increased number of paths generated by CodeQL, which significantly impacts the cost of subsequent contextual analysis, and (2) the higher likelihood of false positives that may result.
>
> To address this concern, we conducted a small-scale experiment on a relatively small project, zt-zip, where both CodeQL and IRIS+GPT4 successfully detected CVE-2018-1002201. Using the **signature-based heuristics** suggested by the reviewer, the pipeline generated 4,441 alarms (compared to 8 for CodeQL and 15 for IRIS+GPT-4), resulting in a False Discovery Rate (FDR) of 98.2%. In contrast, IRIS+GPT4 achieved an FDR of 63.8%, and CodeQL achieved an FDR of 60% on this project. These results demonstrate that the heuristics-based approach yields extremely low precision.
>
> Given that IRIS+GPT-4 outperforms CodeQL on both the number of vulnerabilities detected and the FDR, we maintain that the comparison between IRIS+GPT-4 and CodeQL is fair and reflective of their respective strengths.
>
> > **Q3: “I would like to know the actual cost of LLM call for IRIS on CWE-Bench-Java in total and on average”**
>
> We thank the reviewer for the question. On average, we observe 388.4 LLM calls per project, though this number varies significantly based on the project size. For example, Perwendel Spark, a project with 10K lines of code, required only 43 LLM calls to detect the CVE present within it.
>
> For the final evaluation of IRIS on CWE-Bench-Java, across the 7 evaluated LLMs (including 2 added during the rebuttal) and the 120 projects in our dataset, we made approximately 320,000 LLM calls. While it is challenging to provide an exact cost breakdown, we hope this information offers valuable perspective on the scale and effort involved in our work.

---

> > ### Author Response · Authors · 2024-11-26
> >
> > We appreciate your valuable comments on our paper. We have prepared a rebuttal and tried our best to address your concerns. We are willing to answer any unresolved or further questions that you may have regarding our rebuttal if time is allowed. If our rebuttal has addressed your concerns, we would appreciate it if you would let us know if your final thoughts. Additionally, we will be happy to answer any further questions regarding the paper. Thank you for your time and consideration.

---

### Official Review · Reviewer_EvMB · 2024-11-04

**Soundness:** 2
**Presentation:** 3
**Contribution:** 2
**Rating:** 3
**Confidence:** 4

**Summary:**

This paper presents a hybrid approach that combines static analysis (CodeQL) with LLMs to detect vulnerabilities. Specifically, IRIS uses LLMs to find taint specification of external APIs and use CodeQL to compute paths as context and feed into the prompts to LLMs. The authors evaluated their work on 120 manually  validated Java vulnerabilities. The results show that their approach can detect 56 vulnerabilities, 28 more than the ones reported by CodeQL

**Strengths:**

+ IRIS demonstrated the improved results over CodeQL
+ IRIS found unknown vulnerabilities
+ This paper is well written and contains details.
+ It's interesting to learn that the LLMs can infer taint specifications with 70% accuracy

**Weaknesses:**

Soundness:
This paper contains a problematic evaluation:
- The authors curated 120 examples. All of them are vulnerable labels. The results focus on vulnerabilities detected. The false positive rates are not evaluated?
- There are many vulnerability datasets available, such as PrimeVul, Sven datasets. Why do you not run your tool on other datasets?
- The baselines of CodeQL (QL), Infer (Infer), Spotbugs (SB), and Snyk are all static analysis tools?  Should you also compare any AI based approaches? e.g., what about feeding the entire function into a prompt and see how LLMs perform?
- only 5 models are experimented

Presentation:
- How subset of paths are selected when paths are too long for the prompt? It's not clearly presented
- We typically don't call such an approach neural-symbolic

Contribution:
- This paper makes incremental contributions
(1) Using LLMs for getting taint information. Typically in static analysis, we provide a list of taint APIs, it can be done precisely by the domain experts.
(2) Using paths reported by CodeQL as context when prompting LLM. There have been work that uses output of the static analysis tools like infer as prompts

**Questions:**

-  How are the subset of paths selected when paths are too long?
-  There are many vulnerability datasets available, such as PrimeVul, Sven datasets. Why do you not run your tool on these datasets?

---

> ### Author Response · Authors · 2024-11-23
>
> > **Q1: Comparison of CWE-Bench-Java with existing datasets like PrimeVul and SVEN**
>
> In prior datasets such as PrimeVul and SVEN, the input consists of a single function. In contrast, our task involves analyzing an **entire repository**, which averages 300K lines of code per project. Identifying inter-procedural vulnerability paths within such extensive repositories is akin to finding a "needle in a haystack." Table 5 highlights other key differences between these datasets, underscoring the challenges addressed by CWE-Bench-Java.
>
> From our experience, function-level vulnerability detection is insufficient for tackling real-world vulnerabilities, which typically span multiple functions rather than being isolated to a single one. This perspective is widely shared within the research community [1, 2]. As Reviewer 3b2y also noted, the ability to perform project-level vulnerability detection is a strength of our work.
>
> - [1]: Top Score on the Wrong Exam: On Benchmarking in Machine Learning for Vulnerability Detection, Risse et. al. 2024
> - [2]: Data Quality for Software Vulnerability Datasets, Croft et. al. ICSE 2023
>
> > **Q2: “False positive rates are not evaluated?”**
>
> We do estimate the False Discovery Rate (FDR), which is similar to the false positive rate that the reviewer mentioned but provides a more appropriate measure in our context (Section 3.6). Since we are performing whole-repository analysis, there is no need for an explicit mention of “negative examples” in our dataset—any non-vulnerable paths in the repository naturally serve as negative examples, far outnumbering the vulnerable ones.
>
> > **Q3: “Should you also compare any AI based approaches? e.g., what about feeding the entire function into a prompt and see how LLMs perform?”**
>
> Our dataset expects whole-repository analysis, which is fundamentally different from function-level vulnerability detection, as seen in datasets like PrimeVul and SVEN (Table 5). For this reason, we do not use pure neural approaches as baselines, since incorporating the entire repository (which could have up to 7M lines of code as shown in Table 11) as context for a large language model is computationally intractable.
>
> > **Q4: “How subset of paths are selected when paths are too long for the prompt?”**
>
> We set a hyperparameter $S$ to control the number of intermediate steps in the prompt.For paths with more than $S$ intermediate steps, we divide the path into $S$ equal segments and select one step from each. This selection prioritizes function calls, as they may indicate sanitizations. If no function call is present, we randomly pick one node in the segment. In our experiments, we observe that setting $S$ to 10 provides a good balance between the cost and accuracy so that the prompt contains enough context and would not be too long. We will add this description to the revised version of our paper.
>
> > **Q5: “Typically in static analysis, we provide a list of taint APIs, it can be done precisely by the domain experts”**
>
> As discussed in lines 48-53, it is well-known that existing taint API lists are often insufficient for identifying new, real-world vulnerabilities. These labels are typically created retroactively–after a CVE is discovered, a human expert assigns the corresponding label. This approach introduces significant limitations, as tools relying solely on human-generated labels struggle to keep up with evolving vulnerabilities. Moreover, these labels tend to be rigid and require continuous maintenance to adapt to the dynamic nature of modern software development. In Table 8 in our paper, we present the number of unique APIs presented in our dataset. Manually labeling over thousands of libraries with APIs often appearing in different contexts is prohibitive [3].
>
> - [3]: Scalable Taint Specification Inference with Big Code, Chibotaru et. al. PLDI 2019
>
> > **Q6: “only 5 models are experimented”**
>
> Following the reviewer’s suggestion, we have extended the evaluation with two more popular open-source LLMs, Qwen-2.5-Coder-32B-Instruct and Gemma-2-27B, and we hereby show the high-level performance comparison in the same format as our Table 1. In general, we see that IRIS+LLMs still consistently outperform all traditional baselines. We will incorporate this in the revised version of our paper and we plan to explore more models in the future.
>
> **Table 1 Extended:**
>
> | Method | #Detected | Detection Rate (%) | Avg FDR (%) | Avg F1 Score |
> |------------|-------------|------------|-------|---------------|
> | CodeQL | 27 | 22.50 | 90.03 | 0.076 |
> | IRIS + GPT-4 | 55 | 45.83 | 84.82 | 0.177 |
> | (**NEW**) IRIS + Qwen-2.5-Coder-32B-Instruct | 47 | 39.17 | 92.38 | 0.097 |
> | (**NEW**) IRIS + Gemma-2-27B | 45 | 37.50 | 91.23 | 0.100 |

---

> > ### Author Response · Authors · 2024-11-26
> >
> > We appreciate your valuable comments on our paper. We have prepared a rebuttal with an updated manuscript and tried our best to address your concerns. We are willing to answer any unresolved or further questions that you may have regarding our rebuttal if time is allowed. If our rebuttal has addressed your concerns, we would appreciate it if you would let us know if your final thoughts. Additionally, we will be happy to answer any further questions regarding the paper. Thank you for your time and consideration.

---

### Author Response · Authors · 2024-12-02

As the rebuttal period concludes, we would like to summarize the key improvements and discussions regarding our submission.

Reviewers j2y3, 3b2y, and nTbW unanimously acknowledge the importance and challenge of whole-repository security vulnerability detection. They agree that IRIS, leveraging larger LLMs, significantly outperforms prior methods in both the number of vulnerabilities detected and overall F1 scores. Additionally, all reviewers commend the clarity and detail of our presentation. In response to specific requests for clarification on design decisions (e.g., context window size, types of dataflow nodes, and few-shot/zero-shot strategies), we provided detailed responses during the rebuttal and have incorporated these clarifications into the manuscript.

To address reviewer EvMB's concerns, we clarified that our approach does estimate false positives, as is essential for whole-repository vulnerability detection. Our contextual analysis significantly reduces the false discovery rate. We further clarify that a pure LLM-based approach will face challenges with extremely large repositories (up to 7M lines of code), leading to substantial computational and financial costs.

Regarding reviewer EvMB's comment on the number of LLMs evaluated, we conducted additional experiments during the rebuttal period with two mid-sized LLMs—Gemma-2-27B and Qwen-2.5-Coder-32B-Instruct. Both models outperformed CodeQL in detecting security vulnerabilities, as evidenced by our metrics.

We hope our rebuttal has addressed most of your questions. Please let us know if there are any final comments that we can address.

---

### Meta-Review · Area_Chair_CrDA · 2024-12-21

**Metareview:**

This paper proposes IRIS, a neuro-symbolic approach for security vulnerability detection that integrates large language models with static analysis. Overall, it is a borderline submission with mixed reviews. While most reviewers provide positive feedback, reviewer EvMB holds a differing opinion.

The main concerns raised include:
(1)Problematic evaluation (e.g., all data labeled as vulnerable, absence of a false positive rate);
(2)Lack of experiments on existing datasets;
(3)Comparison with AI-based approaches;
(4)Presentation issues, and
(5)Incremental contribution.

Because reviewer was not heavily involved in the rebuttal discussion, ACr (AC) carefully reviewed all feedback and the rebuttal. The AC agrees with reviewer EvMB's concerns and recognizes the value of all comments. However, after evaluating the rebuttal, the AC believes that the authors have addressed these concerns sufficiently. Therefore, the AC recommends acceptance of this paper, with the expectation that the authors will revise it according to the reviewers’ comments.

**Additional Comments On Reviewer Discussion:**

AC carefully read the reviews and rebuttals. The main concerns from reviewer EvMVare (1) problematic evaluation (e.g., all data are vulnerable label, no false positive rate), (2) no experiment on existing dataset, (3) comparison with AI based approach, (4) presentation problem and (5) incremental contribution. After reading the rebuttal, the AC believes that most of these concerns relate to presentation issues and have been addressed sufficiently. Because this reviewer did not participate in the discussion, the AC places a lower weight on this score.

---

### Decision · Program_Chairs · 2025-01-22

Accept (Poster)